# Numerical Study of the Hygrothermal Effects on Low Velocity Impact Induced Indentation and Its Rebound in Composite Laminate

**Muhammad Yousaf** [1] , **Chuwei Zhou** [1,*], **Yu Yang** [2] **and Li Wang** [1,2]

1   State Key Laboratory of Mechanics and Control of Mechanical Structures, Nanjing University of Aeronautics and Astronautics, Nanjing 210016, China
2   Structural Damage Monitoring Laboratory, Aircraft Strength Research Institute of China, Xi'an 710065, China
*   Correspondence: zcw@nuaa.edu.cn

**Abstract:** Impact indentation is believed to be an effective indication of low-velocity impact (LVI) damage for polymer matrix composites. However, it has been discovered that an indentation can partially rebound over time. Impact indentation and its rebound behavior over a period of time are significantly affected by hygrothermal conditions, especially moisture absorption. Therefore, a good understanding of the moisture-dependent impact indentation and its rebound behavior is helpful for impact damage assessment for composites. In this paper, moisture effects are considered for both the intra-laminar transverse property model and the interlaminar interface model in the simulation of impact indentation. Then, in these two models, viscosities are introduced to represent the indentation rebound over time. In order to validate the proposed models, LVI experiments with different impact energies were conducted on dry and hygrothermal conditioned carbon fiber/epoxy matrix composite laminates. For the specimens, the initial depths of impact dents and their rebounds over time were measured. The specimens of hygrothermal conditions were found with deeper dents compared with dry ones under the same impact energy; and their rebounds were also more significant. These phenomena were explained by the fact that moisture softens epoxy in composite and meanwhile elevates its viscosity. This indentation and its rebound phenomenon were simulated in ABAQUS by considering the moisture effects and viscoelasticity with user-defined material subroutines. These experiments were simulated using the proposed models, and the numerical predictions conformed well with the experimental observations.

**Keywords:** hygrothermal conditioned composite; low-velocity impact dent; rebound of impact dent; moisture-dependent intra-laminate model; moisture-dependent inter-laminate model

## 1. Introduction

Polymer matrix composites (PMCs), especially carbon-fiber-reinforced plastic (CFRP), have the great advantage of having a low specific weight with excellent mechanical properties. Due to this, the use of CFRP has increased widely in many industries, such as the aerospace and automobile sectors. However, CFRP has one big disadvantage too. It has poor properties in the through-thickness direction, which might degrade upon moisture absorption. Thus, making CFRP prone/vulnerable to low-velocity impacts (LVI) and when subjected to different hygrothermal conditions.

When impacted by a low-velocity object, i.e., any falling tool or impactor like a hail stone, composite laminates behave in a relatively brittle manner, resulting in surface and/or internal damage depending upon the absorption of impact energy. Moisture absorption by the composite laminates further increases the damage manifold. Damage may be in the form of material property degradation, delamination (interlaminar damage), matrix/resin cracking, fiber fracture, and matrix/fiber debonding (intralaminar damage).

A dent is a surface trace of an impact event which could relate to all kinds of damage modes.

In order to predict the interlaminar and intralaminar damage induced by LVI, several methodologies are available in the literature. These include strength-based failure methodologies [1–4], fracture-mechanics-based methodologies by using the VCCT technique [5,6] or the CZM technique by modeling cohesive (surface-based or element-based) [7–9] or spring elements [10] at the interface between two adjacent plies, and CDM-based methodologies [11–14].

By briefly summarizing some of these methodologies, Zubillaga presented that matrix cracks are responsible for delamination, and by considering fracture toughness and energy release rate of the interface, he proposed a failure criterion [15]. F. Aymerich, F. Dore, and P. Priolo presented a model to predict delamination induced by LVI in cross ply graphite/epoxy laminated plates using a cohesive interface in their paper [7]. C. Bouvet and S. Rivallant in their paper [16] also discussed that matrix cracking is conventionally the first sign of damage to appear upon impact, followed by delamination, which quickly occurs as damage starts to grow.

CFRP laminates exhibit similar contact force and damage magnitudes under transverse quasi-static and LVI loading [17–20]. Therefore, dynamic impact loading can also be equivalently applied as a quasi-static indentation loading. However, there are some limits where the quasi-static indentation can represent the LVI experiment [21,22].

By discussing indentation further, Karakuzu et al. [23] presented an empirical method to simulate permanent indentation using a numerical study of a glass/epoxy composite plate, which conformed well with the experimental results. In another empirical approach, a study related to indentation and penetration on CFRP laminates was carried out by Caprino et al. [24], which predicted that indentation is an empirical function of impacted energy. He et al. [25] adopted an anisotropic elasto-plasticity theory while considering delamination damage and damping effects for modeling LVI-induced permanent indentation marks on laminated composites.

Surface dents are probably the most direct form of damage observed during a visual check of the impacted site, as the depths of these dents may quantitatively relate to the degree of inside damage. These dents may also rebound depending on the viscoelastic behavior of the composite and the hygrothermal environment during the service life. This might mislead the estimations of impact energy absorbed, as well as the degree of damage during the impact. Therefore, it is necessary to develop a method for the simulation and prediction of LVI damage in terms of dent depth and the rebound of the dent with time in consideration of the environmental conditions.

One of the important properties of PMCs is viscoelasticity, mainly because of the polymer matrix/resin. This means that the dent depth at the impact site is prone to changing over time immediately after the impact. It includes three parts. The first and irreversible part is the impact damage, which causes dislocation of fibers and the matrix. The second part that recovers quickly after the load is removed is the elastic deformation of the whole impacted laminate. The third part, which is responsible for the rebound of the indentation, is the viscoelastic deformation of the resin/matrix of PMCs. It takes much longer for the dent to partially rebound due to viscoelasticity [26]. Furthermore, moisture absorption elevates the viscoelastic material response of composites. These effects are usually not considered in LVI scenarios.

Viscoelastic relations may be expressed in both integral and differential forms, as explained in many published research works. Nima Zobeiry et al. [27] presented a method for FE modeling of isotropic and transversally isotropic viscoelastic materials. Laminae, in a homogenized way, is modeled as transversally isotropic viscoelastic material.

Lin Xiao and Guanhui Wang et al. presented an experimental study related to CFRP subjected to LVI and observed its viscoelastic behavior through the rebound of the indentation dent in terms of depth in post-impact measurements [26]. The relaxation of indentation

depth over time is also reported by Wagih et al. [28] in their paper. A significant relaxation of indentation depth during the first 14 days after the test was observed by them.

The material properties of PMCs degrade severely with hygrothermal conditioning, which subsequently affects their performance in LVI and post-LVI scenarios. Moisture is absorbed in composite laminates through Fickian and/or non-Fickian diffusion.

Thomas [29] showed in his paper that, due to viscoelasticity, the dent depth can decrease over time because of humidity and fatigue. Furthermore, just after the impact, this initial dent depth is three times greater than the final rebounded dent depth in some cases. The decrease in dent depth over time also depends on material properties. Wolff EG et al. [30] explained that due to this moisture-induced matrix plasticization in glass/epoxy composites, the viscoelasticity of the composite could be enhanced.

L.S. Sutherland in his review paper [31] explained that the water absorbed by the composite material may result in degradation of the material properties of the laminate. It severely affects the performance of composites in impact and post-impact scenarios. Berketis et al. [32] observed matrix dissolution and interfacial damage in GRP non-crimp glass/polyester specimens after placing these specimens in a hygrothermal condition (water baths at 65 °C) for up to 30 months. Evidence of water absorption causing interfacial or interlaminar material property degradation in CFRP laminates was also observed by Kimpara and Saito [33]. Strait et al. [34] studied the immersion effects for GRP in synthetic sea water at 60 °C and concluded that moisture-induced matrix plasticization can significantly reduce the impact resistance to failure of GRP composites.

The material properties of FRC degrade severely with hygrothermal aging, which subsequently affects the performance of FRC in LVI scenarios. Parvatareddy et al. [35] observed a 70–75% reduction in the residual strength of FRC during their study on the impact resistance and damage tolerance of FRC exposed at 150 °C. Li et al. [36] studied the hygrothermal aging effects on composites and found that under LVI the damaged area increased, which resulted in a decrease in the post-LVI performance of the composites. Mortas et al. [37] studied the effects of corrosive solutions on composites at elevated temperatures. This type of aging severely affects the residual strength of the FRC. Vieille et al. [38] investigated the effects of moisture and temperature on thermoset (TS) and thermoplastic (TP) resin-based composites.

Castaing P et al. [39] depicted that the glass transition temperature, $T_g$, of epoxy resin/matrix could severely reduce upon moisture absorption. An average of a 20 °C reduction in $T_g$ could occur due to a 1% increment in the absorbed moisture, which is not desirable in a typical glassy epoxy-based polymer. Browning CE et al. [40] found that beyond $T_g$, the physical properties of the epoxy changed rapidly because of structural changes in the epoxy from a glassy (hard) to a rubbery (soft) state. Chateauminois A et al. [41] showed a linear reduction trend in the glass transition temperature $T_g$ with the increase in moisture content as a result of the direct consequence of plasticization of the matrix/resin in glass/epoxy composites.

Acoustic Emission (AE), a non-destructive testing (NDT) technique used in the aerospace industry, represents the transient elastic sound waves that occur when the material undergoes stress. It is used to monitor damage in composite materials [42] by using piezoelectric sensors applied directly on the surface of samples and capturing these elastic waves. Then, an analysis used to distinguish between the damage types (matrix cracks, fiber breaks, fiber/matrix debonding, and delamination) is carried out on the basis of the collected data. This data can be further analyzed to determine different degradation mechanisms in the material during its lifetime. Life cycle analysis (LCA) of the composite materials can be considered via real time damage tracking by AE. Benzeggah and Barre [43], Chen et al. [44], Kim et al. [45], and Kotsikos et al. [46] used AE signals collected during loading to identify damage mechanisms in composite materials. Their studies proposed that the main damage mechanisms in composite materials can be ranked in ascending order based upon AE amplitude signals. While, the main damage mechanisms are matrix

cracking, delamination, debonding, and fiber fracture. Moreover, these AE amplitude signals are dependent on the type of loading and material properties.

In this study, LVI testing of different impact energies (30, 40, and 50 J) is carried out on conditioned and dry specimens to investigate the hygrothermal effects on LVI-induced indentation and its rebound. Then, simulations are performed in ABAQUS with user-defined material subroutines (VUMATs and UMATs). The moisture-dependent intra-laminar viscoelastic model and the moisture-dependent viscoelastic cohesive interface model are implemented through these user-defined material subroutines to demonstrate the initial indentation upon impact and its subsequent rebound over time because of hygrothermal conditioning.

## 2. Moisture-Dependent Viscoelastic Constitution for Composite Laminate

### 2.1. Constitutive Equations for a UD Laminae Ply

Initial ideas are taken from the work carried out by Kaliske and Rothert [47] and, further, from Yousaf and Zhou [48]. The generalized Maxwell rheological model (Figure 1) is used for the derivation/formulation of viscoelastic constitutive equations.

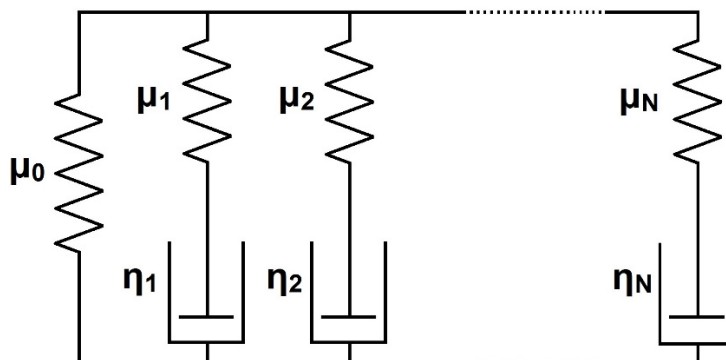

**Figure 1.** Generalized Maxwell rheological model with N Maxwell element chains [48].

For this model, the stress is written as

$$\sigma(t) = \mu_0 \varepsilon(0) + \sum_{i=1}^{N} \mu_i \, exp\left(-\frac{t}{\tau_i}\right) \varepsilon(0) = \Gamma(t) \, \varepsilon(0) \tag{1}$$

Here, $\tau_i = \frac{\eta}{\mu}$ is defined as the relaxation time for each Maxwell element chain.

$$\sigma(t) = \sigma_0(t) + \sum_{i=1}^{N} h_i(t) \tag{2}$$

Here, $\sigma_0(t)$ is the stress in the elastic/Hookean spring element and $h_i(t)$ expresses the stresses inside the Maxwell element chains, i.e.,

$$h_i(t) = \int_0^t \gamma_i \, exp\left(-\frac{t-s}{\tau_i}\right) \frac{\partial \sigma_0(s)}{\partial s} ds \tag{3}$$

Here, $\gamma_i = \frac{\mu_i}{\mu_0}$.

In order to obtain a numerical solution of Equation (3), time-discretized approximation of the 2nd order was implemented in this scheme as

$$h_i^{n+1} = exp\left(-\frac{\Delta t}{\tau_i}\right) h_i^n + \frac{\gamma_i \left(1 - exp\left(-\frac{\Delta t}{\tau_i}\right)\right)}{\frac{\Delta t}{\tau_i}} \left[\sigma_0^{n+1} - \sigma_0^n\right] \tag{4}$$

This indicates a very important result that all the values for $h_i(t)$ are only dependent on the previous values of $h_i$. If these values are known, antecedent values for any given time step can be obtained with the help of the iterative formulation shown in Equation (4). Utilizing the iterative formulation, Equation (2) becomes

$$\sigma^{n+1} = \sigma_0^{n+1} + \sum_{i=1}^{N} h_i^{n+1} \tag{5}$$

The three-dimensional formulation of Equation (5) is

$$\sigma_M^{n+1} = \sigma_{0M}^{n+1} + \sum_{MK} \sum_{p=1}^{N} h_{MKp}^{n+1} \tag{6}$$

Here, the elastic stress is defined as

$$\sigma_{0M}^{n+1} = C_{MK}^e \varepsilon_K^{n+1} \tag{7}$$

Here, the Cauchy stress tensor is defined as

$$\sigma_M = \begin{bmatrix} \sigma_1 \\ \sigma_2 \\ \sigma_3 \\ \sigma_4 \\ \sigma_5 \\ \sigma_6 \end{bmatrix} \tag{8}$$

with $\sigma_{11} = \sigma_1, \sigma_{22} = \sigma_2, \sigma_{33} = \sigma_3, \sigma_{12} = \sigma_{21} = \sigma_4, \sigma_{13} = \sigma_{31} = \sigma_5, \sigma_{23} = \sigma_{32} = \sigma_6$. The strain tensor with Voigt notation can also be written in the same way as

$$\varepsilon_K = \begin{bmatrix} \varepsilon_1 \\ \varepsilon_2 \\ \varepsilon_3 \\ \varepsilon_4 \\ \varepsilon_5 \\ \varepsilon_6 \end{bmatrix} \tag{9}$$

The stiffness tensor is expressed as $C_{MK}^e$. It defines an elastic relationship between stress and strain. This stiffness tensor is used for the transversally isotropic material as

$$C_{MK}^e = \begin{bmatrix} E_1 + 4K_T v_{12}^2 & 2K_T v_{12} & 2K_T v_{12} & 0 & 0 & 0 \\ 2K_T v_{12} & K_T + G_T & K_T - G_T & 0 & 0 & 0 \\ 2K_T v_{12} & K_T - G_T & K_T + G_T & 0 & 0 & 0 \\ 0 & 0 & 0 & G_{12} & 0 & 0 \\ 0 & 0 & 0 & 0 & G_{12} & 0 \\ 0 & 0 & 0 & 0 & 0 & G_T \end{bmatrix} \tag{10}$$

where $K_T = 1/\left(\frac{2(1-v_{23})}{E_2} - \left(\frac{4v_{12}^2}{E_1}\right)\right.$ and $G_T = \frac{E_2}{2(1+v_{23})}$ are the plane strain bulk modulus and transverse shear modulus, respectively [27]. Therefore, the internal stress variables of Equation (4) can be rewritten as

$$h_{MKp}^{n+1} = exp\left(-\frac{\Delta t}{\tau_{MKp}}\right) h_{MKp}^n + \frac{\gamma_{MKp}\left(1 - exp\left(-\frac{\Delta t}{\tau_{MKp}}\right)\right)}{\frac{\Delta t}{\tau_{MKp}}} \left[C_{MK}^e \varepsilon_K^{n+1} - C_{MK}^e \varepsilon_K^n\right] \tag{11}$$

Now the moisture concentration variables $G_0$ and $G_1$ are added in Equation (11), where $G_0$ indicates the increase or decrease in the instantaneous stiffness and $G_1$ indicates the increase or decrease in viscoelastic/transient stiffness with the change in moisture concentration. Here,

$$G_0 = \frac{\left( C_{MK}^e - \alpha \times \frac{c}{C_0} \right)}{C_{MK}^e} \tag{12}$$

and

$$G_1 = 1 - \beta \times c/C_0 \tag{13}$$

where $C_0$ is the applied moisture concentration (ppm or gm/m$^3$) that corresponds to the relative humidity level, $\alpha$ (GPa) is the slope of the graph of instantaneous modulus $C_{MK}^e$ (GPa) versus non-dimensional moisture concentration $c/C_0$ with a value of $c$ (ppm or gm/m$^3$) which corresponds to a given material point coordinate and time, and $\beta$ is a constant.

Now Equation (11) can be written as

$$h_{MKp}^{n+1} = exp\left( -\frac{\Delta t}{\tau_{MKp}} \right) G_1 h_{MKp}^n + \frac{G_1 \gamma_{MKp} \left( 1 - exp\left( -\frac{\Delta t}{\tau_{MKp}} \right) \right)}{\frac{\Delta t}{\tau_{MKp}}} \left[ G_0 C_{MK}^e \varepsilon_K^{n+1} - G_0 C_{MK}^e \varepsilon_K^n \right] \tag{14}$$

After expanding Equation (6), the Cauchy stress in incremental formulation type is written as

$$\sigma_M^{n+1} = G_0 C_{MK}^e \varepsilon_K^{n+1} + \sum_{MK} \sum_{p=1}^N exp\left( -\frac{\Delta t}{\tau_{MKp}} \right) G_1 h_{MKp}^n + \frac{G_1 \gamma_{MKp} \tau_{MKp} \left( 1 - exp\left( -\frac{\Delta t}{\tau_{MKp}} \right) \right)}{\Delta t} \left[ G_0 C_{MK}^e \varepsilon_K^{n+1} - G_0 C_{MK}^e \varepsilon_K^n \right] \tag{15}$$

This is an iterative formulation for the stress in the viscoelastic material and it is used in the ABAQUS user material subroutine of UMAT.

In this paper, viscoelastic UD Laminae Ply is simulated by considering only two Maxwell element chains in parallel with the Hookean springs of all the members of the stiffness tensor. Therefore, the constitutive Equation (15) takes the shape of

$$\sigma_M^{n+1} = G_0 C_{MK}^e \varepsilon_K^{n+1} + exp\left( -\frac{\Delta t}{\tau_{MK1}} \right) G_1 h_{MK1}^n + exp\left( -\frac{\Delta t}{\tau_{MK2}} \right) G_1 h_{MK2}^n +$$

$$\left\{ 1 + \frac{G_1 \gamma_{MK1} \tau_{MK1} \left( 1 - exp\left( -\frac{\Delta t}{\tau_{MK1}} \right) \right)}{\Delta t} + \frac{G_1 \gamma_{MK2} \tau_{MK2} \left( 1 - exp\left( -\frac{\Delta t}{\tau_{MK2}} \right) \right)}{\Delta t} \right\} \left[ G_0 C_{MK}^e \Delta \varepsilon_K \right] \tag{16}$$

where

$$\Delta \varepsilon_K = \varepsilon_K^{n+1} - \varepsilon_K^n \tag{17}$$

The internal stresses in the Maxwell element chains are updated by using the relationship defined by Equation (14) in each iteration.

Constitutive Equation (16) is further used to calculate the tangent modulus/Jacobian matrix, which is the slope for the stress–strain curve

$$\mathbb{C}_{MK}^{n+1} = \frac{\partial \Delta \sigma}{\partial \Delta \varepsilon} = \left\{ 1 + \sum_{MK} \sum_{p=1}^2 \frac{G_1 \gamma_{MKp} \left( 1 - exp\left( -\frac{\Delta t}{\tau_{MKp}} \right) \right)}{\frac{\Delta t}{\tau_{MKp}}} \right\} \left[ G_0 C_{MK}^{e\ n+1} \right] \tag{18}$$

where $C_{MK}^e$ is the stiffness matrix defined in Equation (10).

### 2.2. Failure Law of UD Laminae Ply

The in-plane 3D Hashin Failure Criterion is used to simulate lamina-ply level failures, i.e., matrix failure and fiber failure (Table 1), to simulate impact damage during LVI

testing. By adding a moisture concentration variable $(G_0)_{Impact}$ in the formulation, the instantaneous stiffness $E_0$ changes with moisture concentration.

$$(G_0)_{Impact} = \frac{\left(E_0 - \alpha \times \frac{c}{C_0}\right)}{E_0} \tag{19}$$

**Table 1.** Damage Criterion for UD Laminae [49].

| Damage Type | Failure Mode | | Damage Initiation |
|---|---|---|---|
| | **Matrix** | Tension Cracking | $(\sigma_{yy}/Y_T)^2 + (\tau_{xy}/S_{xy})^2 + (\tau_{yz}/S_{yz})^2 \geq 1$ |
| **Lamina Ply Level** | | Compression Cracking | $(\sigma_{yy}/Y_C)^2 + (\tau_{xy}/S_{xy})^2 + (\tau_{yz}/S_{yz})^2 \geq 1$ |
| | **Fiber** | Tension Failure | $(\sigma_{xx}/X_T)^2 + (\tau_{xy}/S_{xy})^2 + (\tau_{yz}/S_{yz})^2 \geq 1$ |
| | | Compression Failure | $(\sigma_{xx}/X_C)^2 \geq 1$ |

Where $\sigma_{xx}$ and $\sigma_{yy}$ are in plane stresses in fiber and transverse directions. $\tau_{xy}$ and $\tau_{yz}$ are shear stresses. $X_T$ and $X_C$ are fiber tension and compression strength. $Y_T$ and $Y_C$ are matrix tension and compression strength. $S_{xy}$ and $S_{yz}$ are shear strength, respectively.

*2.3. Constitutive Equations for the Cohesive Interface*

A similar approach was extended to formulate constitutive equations for the viscoelastic cohesive interface for initiation, evolution of damage, and rebound of deformation in an impacted laminate. To derive constitutive equations/formulations for the viscoelastic cohesive interface, let us introduce 3D formulation, vector notation adaptation in Equation (5), i.e.,

$$t_M^{n+1} = t_{0M}^{n+1} + \sum_{MK} \sum_{p=1}^{N} h_{MKp}^{n+1} \tag{20}$$

Here, the elastic stress is defined as

$$t_{0M}^{n+1} = E_{MK}^e \varepsilon_K^{n+1} \tag{21}$$

Here, the traction stress vector is defined as

$$t_M = \begin{bmatrix} t_1 \\ t_2 \\ t_3 \end{bmatrix} \tag{22}$$

with the subscripts of 1, 2, and 3 corresponding to normal and two shear directions in the interface. The nominal strain vector can also be written as

$$\varepsilon_K = \begin{bmatrix} \varepsilon_1 \\ \varepsilon_2 \\ \varepsilon_3 \end{bmatrix} \tag{23}$$

The elasticity vector is expressed as $E_{MK}^e$. It defines an elastic relationship between stress and strain. This elasticity matrix is used for the isotropic material as

$$E_{MK}^e = \begin{bmatrix} K_N(1-d) & 0 & 0 \\ 0 & K_S(1-d) & 0 \\ 0 & 0 & K_T(1-d) \end{bmatrix} \tag{24}$$

where $K_N, K_S, K_T, d$ are the elastic stiffness values in normal and in two shear directions and the damage variable, respectively. Therefore, the internal stress variables of Equation (4) can be rewritten as

$$h_{MKp}^{n+1} = exp\left(-\frac{\Delta t}{\tau_{MKp}}\right) h_{MKp}^n + \frac{\gamma_{MKp}\left(1 - exp\left(-\frac{\Delta t}{\tau_{MKp}}\right)\right)}{\frac{\Delta t}{\tau_{MKp}}} \left[E_{MK}^e \varepsilon_K^{n+1} - E_{MK}^e \varepsilon_K^n\right] \quad (25)$$

Now, the moisture concentration variables $G_0$ and $G_1$ are added in Equation (25), where $G_0$ indicates the increase or decrease in the instantaneous stiffness and $G_1$ indicates the increase or decrease in viscoelastic/transient stiffness with the change in moisture concentration. Here,

$$G_0 = \frac{\left(E_{MK}^e - \alpha \times \frac{c}{C_0}\right)}{E_{MK}^e} \quad (26)$$

and

$$G_1 = 1 - \beta \times c/C_0 \quad (27)$$

where $C_0$ is the applied moisture concentration (ppm or gm/m$^3$) that corresponds to the relative humidity level, $\alpha$ (GPa/mm) is the slope of the graph of the instantaneous modulus $E_{MK}^e$ (GPa/mm) versus the non-dimensional moisture concentration $c/C_0$ with a value of c (ppm or gm/m$^3$) that corresponds to a given material point coordinate and time, and $\beta$ is a constant.

Now, Equation (25) can be written as

$$h_{MKp}^{n+1} = exp\left(-\frac{\Delta t}{\tau_{MKp}}\right) G_1 h_{MKp}^n + \frac{G_1\gamma_{MKp}\left(1 - exp\left(-\frac{\Delta t}{\tau_{MKp}}\right)\right)}{\frac{\Delta t}{\tau_{MKp}}} \left[G_0 E_{MK}^e \varepsilon_K^{n+1} - G_0 E_{MK}^e \varepsilon_K^n\right] \quad (28)$$

After expanding Equation (20), the traction stress vector in the incremental formulation type is written as

$$t_M^{n+1} = G_0 E_{MK}^e \varepsilon_K^{n+1} + \sum_{MK}\sum_{p=1}^N exp\left(-\frac{\Delta t}{\tau_{MKp}}\right) G_1 h_{MKp}^n + \frac{G_1\gamma_{MKp}\tau_{MKp}\left(1 - exp\left(-\frac{\Delta t}{\tau_{MKp}}\right)\right)}{\Delta t} \left[G_0 E_{MK}^e \varepsilon_K^{n+1} - G_0 E_{MK}^e \varepsilon_K^n\right] \quad (29)$$

This is an iterative formulation for the stress in a viscoelastic interface and it was also implemented in the UMAT subroutine.

In this paper, the viscoelasticity of interface was also simulated by two Maxwell element chains in parallel with the Hookean springs of all the members of the elasticity vector. Multiple values of bulk modulus and shear modulus were applied in normal and shear directions of the cohesive interface. Therefore, the constitutive Equation (29) takes the shape of

$$t_M^{n+1} = G_0 E_{MK}^e \varepsilon_K^{n+1} + exp\left(-\frac{\Delta t}{\tau_{MK1}}\right) G_1 h_{MK1}^n + exp\left(-\frac{\Delta t}{\tau_{MK2}}\right) G_1 h_{MK2}^n$$

$$+ \left\{1 + \frac{G_1\gamma_{MK1}\tau_{MK1}\left(1 - exp\left(-\frac{\Delta t}{\tau_{MK1}}\right)\right)}{\Delta t} + \frac{G_1\gamma_{MK2}\tau_{MK2}\left(1 - exp\left(-\frac{\Delta t}{\tau_{MK2}}\right)\right)}{\Delta t}\right\} \left[G_0 E_{MK}^e \Delta\varepsilon_K\right] \quad (30)$$

where

$$\Delta\varepsilon_K = \varepsilon_K^{n+1} - \varepsilon_K^n \quad (31)$$

The internal stresses in the Maxwell element chains are updated by using the relationship defined by Equation (28) in each iteration.

The constitutive Equation (30) is further used to calculate the tangent modulus/Jacobian matrix, which is the slope for the stress–strain curve

$$\mathbb{E}_{MK}^{n+1} = \frac{\partial \Delta t}{\partial \Delta \varepsilon} = \left\{ 1 + \sum_{MK} \sum_{p=1}^{2} \frac{G_1 \gamma_{MKp} \left( 1 - exp\left( -\frac{\Delta t}{\tau_{MKp}} \right) \right)}{\frac{\Delta t}{\tau_{MKp}}} \right\} \left[ G_0 E_{MK}^{en+1} \right] \quad (32)$$

where $E_{MK}^{e}$ is the stiffness matrix defined in Equation (24).

### 2.4. Damage Law of Cohesive Interface

The stress-based quadratic traction–separation failure criterion and linear softening mixed mode B-K law are used to simulate cohesive interface failure initiation and propagation, respectively (Table 2 and Figure 2). The instantaneous elasticity $K_{M0}$ changes with the change in moisture concentration.

$$(G_0)_{Impact} = \frac{\left( K_{M0} - \alpha \times \frac{c}{C_0} \right)}{K_{M0}} \quad (33)$$

where $t_n$, $t_s$, and $t_t$ are interface stresses in normal and two shear directions, respectively. $T_n^f$ is the interface strength in the normal direction and $T_i^f$ is the interface strength in two shear directions. $\delta_n$ or $\delta_3$, $\delta_s$, $\delta_t$ are pure mode displacement values in normal and two shear directions, respectively. For the mixed mode, $\delta_m$, $\delta_m^0$, $\delta_m^f$ represent the effective displacement value, effective displacement for propagation onset, and effective displacement for total failure. Whereas $d$ and $\alpha$ are variables for damage evolution in terms of delamination and maximum displacement reached historically (a state variable), respectively. $K_M$ is the mixed mode elastic stiffness value.

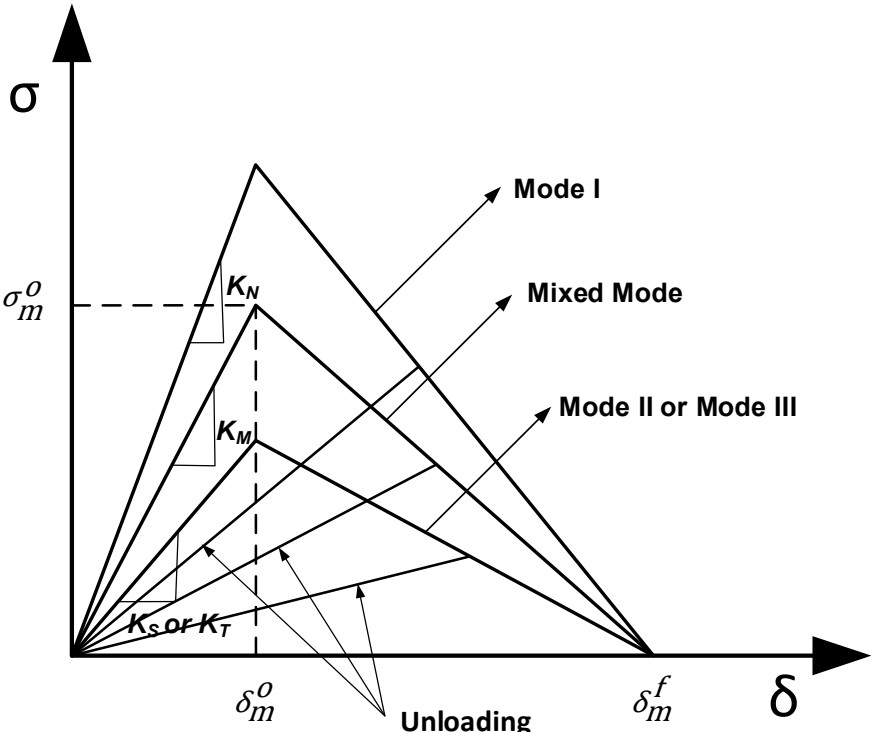

**Figure 2.** Schematic Diagram of the Cohesive Interface Linear Softening Model [50,51].

**Table 2.** Damage Criterion for the Cohesive Interface [50,51].

| Damage Type | Damage Initiation | Damage Propagation |
|---|---|---|
| Cohesive Layer Interface | $\left(\langle t_n \rangle / T_n^f \right)^2 + \left( t_s / T_i^f \right)^2 + \left( t_t / T_i^f \right)^2 \geq 1$ <br><br> $\delta_m = \sqrt{\langle \delta_n^2 \rangle + \delta_s^2 + \delta_t^2}$ <br><br> $d = \delta_m^f (\alpha - \delta_m^0) / \alpha \left( \delta_m^f - \delta_m^0 \right)$ | Linear Softening Mixed Mode B-K Law <br> $\sigma = D \cdot \delta$ <br><br> $D_{ij} = \begin{cases} K_M \overline{\delta}_{ij}, & \alpha \leq \delta_m^0 \\ \overline{\delta}_{ij}[(1-d)K_M + dK_M \frac{\langle -\delta_3 \rangle}{-\delta_3} \overline{\delta}_{i3}], & \delta_m^0 < \alpha < \delta_m^f \\ \overline{\delta}_{i3} \overline{\delta}_{3j} \frac{\langle -\delta_3 \rangle}{-\delta_3} K_M, & \alpha \geq \delta_m^f \end{cases}$ |

## 3. Experiment and FEM Simulation

### 3.1. Experimental Methodology

Specimens were made of T700/QY9510 composite. The supplier of these composite laminates was "Xie Chuang Composite Material Co., Ltd. Dongguan, China." The supplier manufactured these in the form of prepregs using a hot-pressing process. The fibers used in these prepregs were carbon fiber HF30 (with properties similar to T700) and the matrix/resin used had properties similar to epoxy EM817. Each specimen had dimensions of 150 mm × 100 mm × 4.97 mm (the thickness of single ply is 0.155 mm approximately) and had a ply-pattern of [45/0/-45/90]$_{4S}$. A total of 33 such specimens were used.

Pre-conditioning:

All these specimens were heated in an oven at a temperature of 40 °C for 1 h in order to eliminate any moisture before being subjected to different hygrothermal conditions.

After drying, the initial weight of these specimens was measured on a digital mass balance with a precision of up to four decimal points.

Conditioning:

Hygrothermal aging on 24 specimens was completed by selecting the following conditions in an environment conditioning chamber:

I.　25 °C/RH: 85%;
II.　25 °C/RH: 100%.

To determine the moisture absorption by each specimen, the weight gain was measured on an experimental lab ledger at regular time intervals. The percentage weight gain was calculated for each specimen until saturation in accordance with the ASTM D5529/D5229M-92 [52] standard. Initially, weight measurements were undertaken in shorter time intervals of 24–48 h, and later measurement intervals were 1–2 weeks. Six specimens were used to calculate the maximum moisture gain percentage and saturation time for each hygrothermal condition.

The composite material under study here followed Fick's Diffusion Law in moisture absorption behavior, as indicated by an initial linear increase in the moisture level, thereby steadily trending towards saturation.

$$\text{Fick's Diffusion Law}: \ D = \pi \left( \frac{h}{4M_s} \right)^2 \left( \frac{M_2 - M_1}{\sqrt{t_2} - \sqrt{t_1}} \right)^2 = \pi \left( \frac{h}{4M_s} \right)^2 k^2 \qquad (34)$$

where $h$ is the thickness of the composite, $M_s$ is the maximum moisture uptake in percentage at the saturation time, $M_1$ and $M_2$ are moisture gain values in percentage at time $t_1$ and $t_2$, respectively. $k$ is the slope of the linear part of the $M$ versus $\sqrt{t}$ curve.

A moisture absorption curve can be numerically fitted using Equation (35). Only moisture diffusion along the thickness direction, or z-direction, was considered. Moisture concentrations were applied on both sides, i.e., at thickness t = 0 mm and at thickness t = 5.022 mm, as shown in Figure 3. Other sides/edges of the specimen were coated with sealing paint to stop moisture diffusion. Thus, reducing the case to a one-dimensional moisture diffusion problem. Figure 4 shows the maximum moisture gains of 0.32% and 0.37% at the saturation time for conditioned cases I and II, respectively. From Figure 4,

using slope $k$, the moisture diffusion value of $D_z = 3.64 \times 10^{-7}$ mm$^2$/s was calculated along the thickness direction from Equation (34) at a 25 °C temperature and 1 bar pressure.

$$\text{Fickian Numerical Fit}: M = M_s \left[ 1 - exp \left\{ -7.3 \left( \frac{D \cdot t}{h^2} \right)^{0.75} \right\} \right] \tag{35}$$

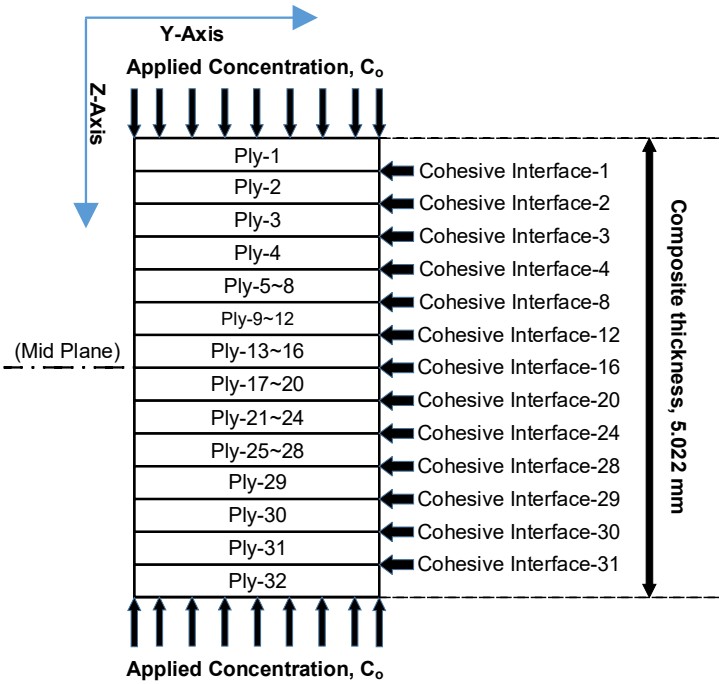

**Figure 3.** Moisture Diffusion Analysis—Schematic for the Applied Concentration (along the thickness or z-direction).

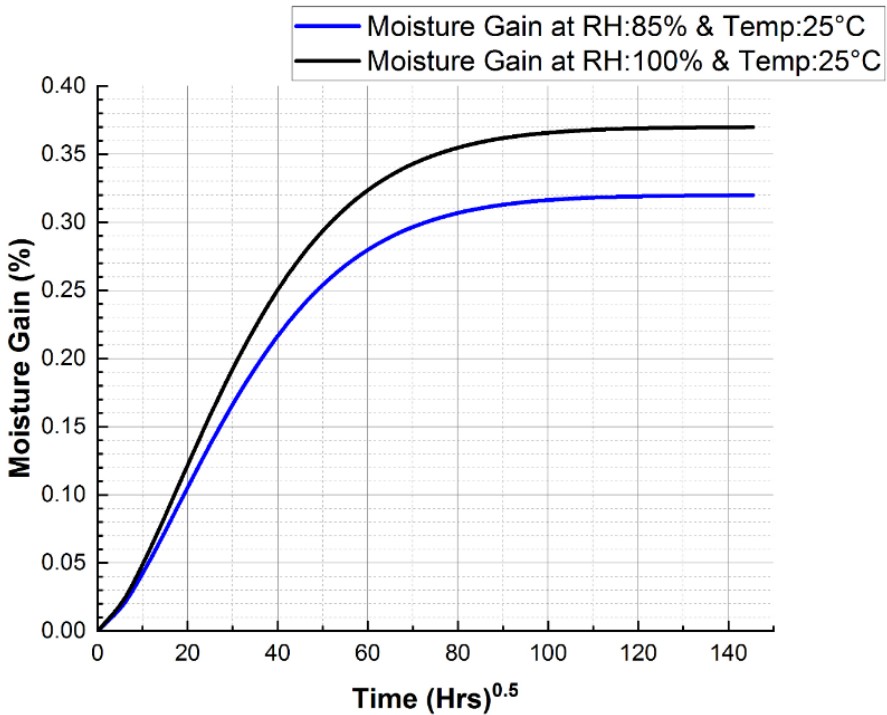

**Figure 4.** Moisture Gain vs. Time Curve.

Impact testing and dent rebound measurements:

Then, these dry (at 25 °C) and conditioned specimens were subjected to LVI testing by using an impact machine in accordance with the ASTM D7136M-15 standard [53] to determine the moisture absorption effects in terms of an LVI-induced dent and its rebound phenomenon over time.

The impactor is made of steel and is hemispherical, with a mass of 5 kg and a diameter of 16 mm.

Unidirectional (UD) laminae ply has transversally isotropic material behavior. The material properties of UD lamina ply are presented in Table 3.

**Table 3.** Material Properties of a single Unidirectional (UD) Ply.

| Elastic Constants of a Single UD Ply | Strength of a Single UD Ply |
| --- | --- |
| $E_1$ = 144.62 GPa | $X_T$ = 2612.24 MPa |
| $E_2$ = 9.76 GPa | $X_C$ = 1583.47 MPa |
| $G_{12}$ = 5.44 GPa | $Y_T$ = 58.25 MPa |
| $G_{23}$ = 3.92 GPa | $Y_C$ = 161.76 MPa |
| $v_{12}$ = 0.31 | $S_{12}$ = 126.79 MPa |
| $v_{23}$ = 0.46 | $S_{23}$ = 91.84 MPa |

The material properties of the cohesive interface are provided in Table 4.

**Table 4.** Material Properties of the Cohesive Interface between Plies.

| $K_n$ (GPa/mm) | $K_i$ (GPa/mm) | $T_n^f$ (MPa) | $T_i^f$ (MPa) | $\delta_n^f$ (mm) | $\delta_n^f$ (mm) | $E_{nb}$ (GPa/mm) | $\delta_n^r$ (mm) |
| --- | --- | --- | --- | --- | --- | --- | --- |
| 1390.0 | 510.0 | 65.5 | 95.5 | 0.014 | 0.025 | 26.5 | 0.043 |

The elasticities and strengths of the specimens presented in Tables 3 and 4 were offered by the supplier "Xie Chuang Composite Material Co., Ltd., Dongguan, China".

Each of the three specimens was used in 30, 40, and 50 J impact energy cases for dry and conditioned specimens. Then, averaged values were used to calculate the initial indentation depths and their recovery/rebound with a stipulated period of time in hours. Depth was measured with a depth screw gauge/micrometer.

Dent depth rebound data with time against each impact energy case for dry specimens are presented in Table 5.

**Table 5.** Experimental Results of Indentation Depth Rebound with time (Dry Specimen at 25 °C) [47].

| Dry Specimen at 25 °C Temperature | | | | | | |
| --- | --- | --- | --- | --- | --- | --- |
| Case-I: 50J | | Case-II: 40J | | Case-III: 30J | | |
| Time (h) | Depth (mm) | Time (h) | Depth (mm) | Time (h) | Depth (mm) | |
| 0 | 0.348 | 0 | 0.238 | 0 | 0.218 | $(d_i)_{Dry}$ |
| 0.75 | 0.338 | 0.5 | 0.233 | 0.5 | 0.211 | |
| 4 | 0.311 | 1 | 0.229 | 1.5 | 0.209 | |
| 16 | 0.288 | 4 | 0.226 | 6 | 0.204 | |
| 24 | 0.282 | 14 | 0.221 | 18.33 | 0.198 | |
| 48 | 0.273 | 24 | 0.218 | 50.5 | 0.191 | |
| 76 | 0.267 | 42.5 | 0.214 | 77.5 | 0.187 | |

**Table 5.** *Cont.*

| Dry Specimen at 25 °C Temperature | | | | | | |
|---|---|---|---|---|---|---|
| **Case-I: 50J** | | **Case-II: 40J** | | **Case-III: 30J** | | |
| **Time (h)** | **Depth (mm)** | **Time (h)** | **Depth (mm)** | **Time (h)** | **Depth (mm)** | |
| 100 | 0.264 | 51 | 0.213 | 101.33 | 0.185 | |
| 124 | 0.262 | 72 | 0.210 | 125.5 | 0.183 | |
| 148 | 0.261 | 98 | 0.208 | 150 | 0.182 | |
| 172 | 0.261 | 125 | 0.206 | 174 | 0.182 | |
| 196 | 0.260 | 150 | 0.205 | 198.33 | 0.181 | |
| | | 174 | 0.204 | | | |
| 220 | 0.260 | 198 | 0.204 | 222.33 | 0.181 | |
| | | 222 | 0.204 | | | |
| 242 | 0.260 | 244 | 0.204 | 244.33 | 0.181 | $(d_f)_{Dry}$ |

The first and the last lines in Table 5 show the initial and the final indentation depths. Tables 6 and 7 list dent depth rebound data with time against each impact energy case for specimens at two relative humidity levels.

**Table 6.** Experimental Results of the Indentation Depth Rebound with time (Specimen at 25 °C/RH:85%).

| At 25 °C Temperature and RH: 85% | | | | | | |
|---|---|---|---|---|---|---|
| **Case-I: 50J** | | **Case-II: 40J** | | **Case-III: 30J** | | |
| **Time (h)** | **Depth (mm)** | **Time (h)** | **Depth (mm)** | **Time (h)** | **Depth (mm)** | |
| 0 | 0.351 | 0 | 0.242 | 0 | 0.220 | $(d_i)_{RH:85\%}$ |
| 1 | 0.329 | 1 | 0.238 | 1 | 0.212 | |
| 3 | 0.310 | 2 | 0.230 | 2 | 0.201 | |
| 16 | 0.279 | 4 | 0.225 | 4 | 0.194 | |
| 24 | 0.268 | 18 | 0.214 | 18 | 0.182 | |
| 48 | 0.249 | 24 | 0.211 | 40 | 0.171 | |
| 76 | 0.238 | 48 | 0.201 | 72 | 0.164 | |
| 100 | 0.232 | 72 | 0.192 | 108 | 0.157 | |
| 124 | 0.226 | 96 | 0.185 | 140 | 0.154 | |
| 148 | 0.221 | 120 | 0.179 | 166 | 0.151 | |
| 172 | 0.219 | 140 | 0.174 | 196 | 0.15 | |
| 196 | 0.217 | 172 | 0.171 | 220 | 0.149 | |
| 220 | 0.217 | 198 | 0.168 | | | |
| 244 | 0.217 | | | 248 | 0.149 | |
| 268 | 0.217 | 220 | 0.168 | | | |
| 280 | 0.217 | 254 | 0.168 | 260 | 0.149 | $(d_f)_{RH:85\%}$ |

**Table 7.** Experimental Results of the Indentation Depth Rebound with time (Specimen at 25 °C/RH:100%).

| At 25 °C Temperature and RH: 100% | | | | | | |
|---|---|---|---|---|---|---|
| **Case-I: 50J** | | **Case-II: 40J** | | **Case-III: 30J** | | |
| **Time (h)** | **Depth (mm)** | **Time (h)** | **Depth (mm)** | **Time (h)** | **Depth (mm)** | |
| 0 | 0.353 | 0 | 0.244 | 0 | 0.221 | $(d_i)_{RH:100\%}$ |
| 1 | 0.325 | 1 | 0.241 | 1 | 0.209 | |
| 3 | 0.312 | 2 | 0.235 | 2 | 0.198 | |
| 16 | 0.276 | 4 | 0.229 | 4 | 0.192 | |
| 24 | 0.262 | 16 | 0.216 | 24 | 0.174 | |
| 48 | 0.231 | 24 | 0.206 | 50.5 | 0.158 | |
| 76 | 0.217 | 48 | 0.188 | 72 | 0.151 | |
| 98 | 0.212 | 72 | 0.177 | 108 | 0.145 | |
| 122 | 0.207 | 96 | 0.169 | 132 | 0.143 | |
| 148.5 | 0.203 | 120 | 0.163 | 166 | 0.142 | |
| 170.5 | 0.203 | 148 | 0.159 | 190.5 | 0.142 | |
| 198 | 0.201 | 172 | 0.157 | 224 | 0.141 | |
| 222 | 0.201 | 196 | 0.157 | | | |
| 248 | 0.201 | 220 | 0.156 | 248 | 0.141 | |
| 272 | 0.201 | 248 | 0.156 | | | |
| 296 | 0.201 | 272 | 0.156 | 271.5 | 0.141 | $(d_f)_{RH:100\%}$ |

Similar to Table 5, the first and last lines in these two tables show the initial and final indentation depths for the specimens in two hygrothermal conditions.

The graphical representation of dent depth rebound data with time is shown in Figure 5.

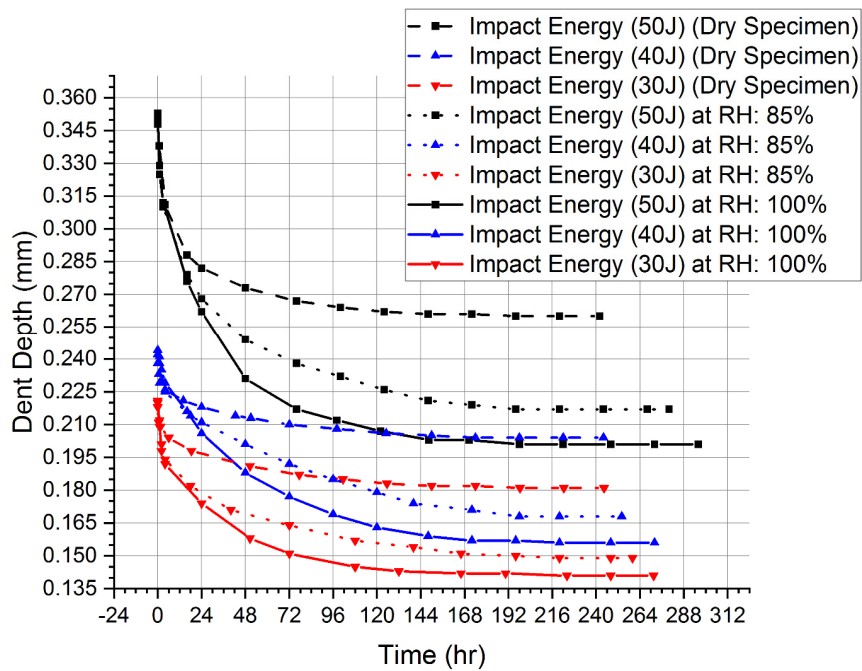

**Figure 5.** Indentation Rebound: Depth vs. Time Graph.

Figure 6 shows a 40J dry impacted specimen from the impacted side and back side. The back side surface of the specimen remained undamaged.

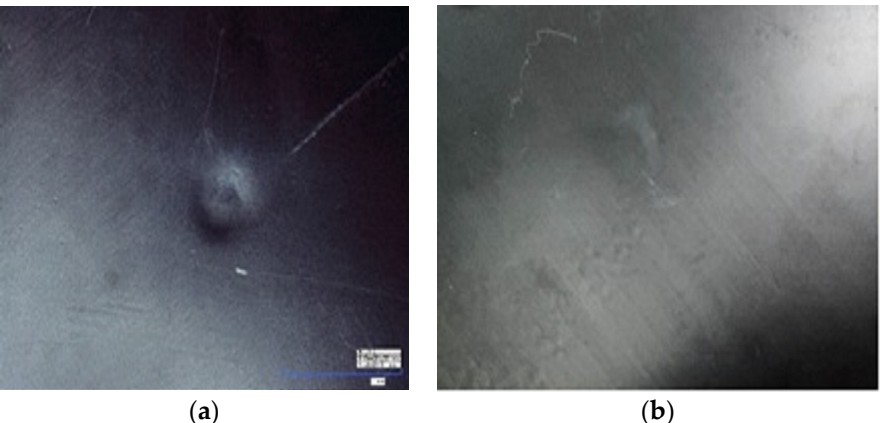

(a)                                                                        (b)

**Figure 6.** Images of the 40J Impact Case (Dry Specimen at 25 °C) [48]. (**a**) View from the Impact Side. (**b**) View from the Back Side.

By taking the data points of dent depths for the 50, 40, and 30J impact cases for dry specimens, the following two-exponential decay type equation (Equation (36)) was used in the curve fitting technique, which helped with dent depth calculation in (mm) units for a given time in (hrs) units, as shown in Figure 7:

$$y = A1 \times exp\left(-\frac{x}{t1}\right) + A2 \times exp\left(-\frac{x}{t2}\right) + y0 \tag{36}$$

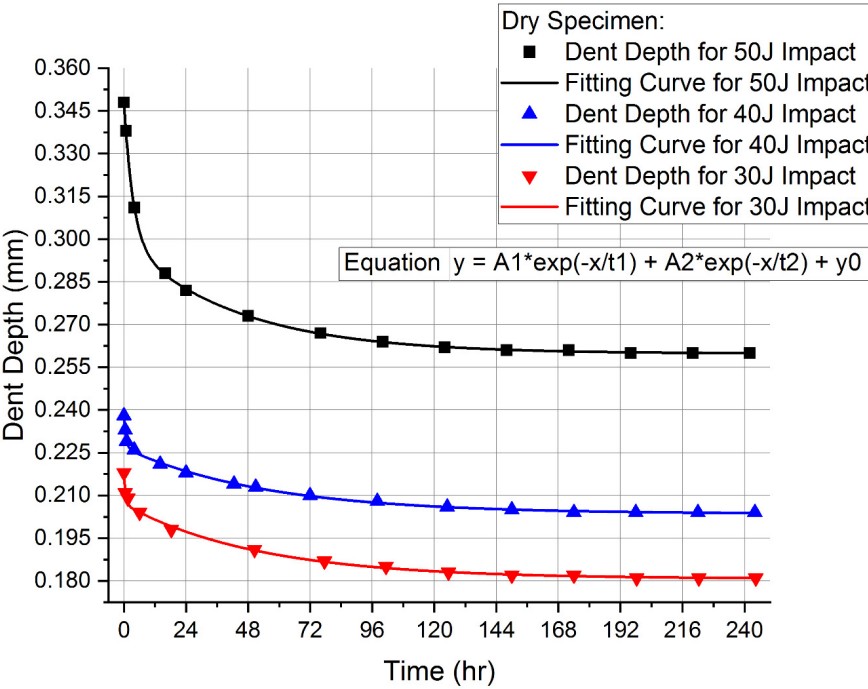

**Figure 7.** Fitting Curves for 30, 40, and 50J Impact Cases—Dent Depth vs. Time Graph (Dry Specimen) [48].

Similarly, Equation (36) was also acquired in the curve fitting technique for dent depth calculation in (mm) units for a given time in (hrs) units for 50, 40, and 30J impact cases for the conditioned specimens.

The values for the variables used in Equation (36) for each impact energy case for dry and conditioned specimens are presented in Table 8.

**Table 8.** Variables Values of the Fitting Curve Equation for 30, 40, and 50J Impact Cases (Dry and Conditioned Specimens).

| Hygrothermal Condition | Impact Energy | A1 | t1 | A2 | t2 | y0 | Adj. R-Square |
|---|---|---|---|---|---|---|---|
| 25 °C, Dry Specimen [47] | 50J Case | 0.04918 | 3.48385 | 0.03910 | 44.22815 | 0.25979 | 0.99989 |
| | 40J Case | 0.01135 | 0.78406 | 0.02320 | 54.84134 | 0.20355 | 0.99823 |
| | 30J Case | 0.01074 | 0.74417 | 0.02614 | 52.47742 | 0.18074 | 0.99644 |
| 25 °C, RH: 85% | 50J Case | 0.05212 | 2.38215 | 0.08222 | 56.32780 | 0.21584 | 0.99888 |
| | 40J Case | 0.01713 | 2.18399 | 0.06377 | 92.10215 | 0.16190 | 0.99726 |
| | 30J Case | 0.02978 | 2.37639 | 0.04347 | 70.98319 | 0.14747 | 0.99808 |
| 25 °C, RH: 100% | 50J Case | 0.03347 | 0.76271 | 0.11800 | 36.08439 | 0.20149 | 0.99932 |
| | 40J Case | 0.00987 | 1.92337 | 0.08031 | 55.13927 | 0.15452 | 0.99905 |
| | 30J Case | 0.02600 | 1.44405 | 0.05480 | 44.42592 | 0.14066 | 0.99880 |

### 3.2. Simulation Methodology

To simulate the indentation depth and its rebound phenomenon in a composite plate, the following methodology was adopted:

The simulation was divided into three parts, as shown in Figure 8.

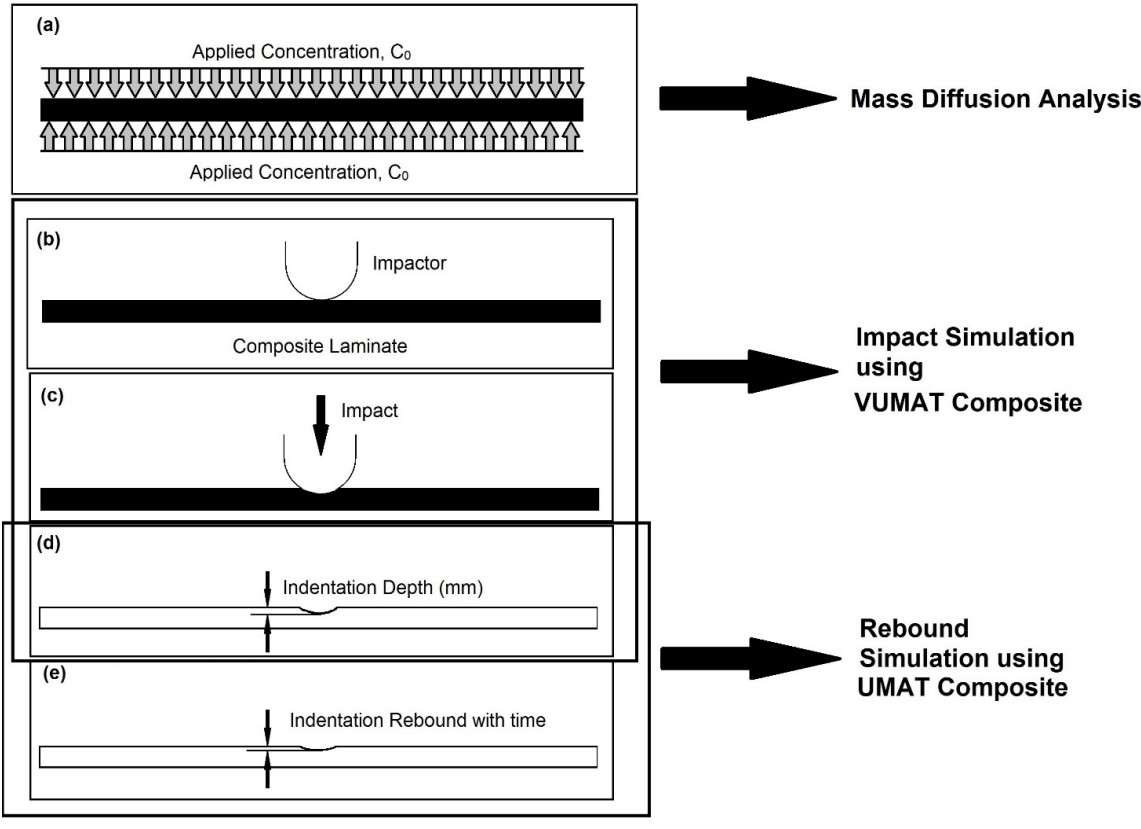

**Figure 8.** Simulation Methodology: Moisture diffusion followed by LVI-induced indentation and the rebound with time: (**a**) moisture diffusion; (**b**) before impact; (**c**) impact dent; (**d**) initial dent depth; (**e**) final dent depth.

Firstly, the moisture diffusion of specimen was simulated using mass diffusion analysis in ABAQUS Standard by applying different concentrations relative to different humidity levels. Additionally, a moisture concentration curve along the thickness direction (z-direction) was obtained. The degradation of instantaneous modulus and/or transient/viscoelastic modulus due to moisture concentration was incorporated in constitutive material models and implemented via user-defined material subroutines, i.e., VUMAT and UMAT.

Secondly, the impacts at different energy levels (30, 40, and 50 J) on dry and conditioned specimens were simulated in ABAQUS Explicit as a dynamic modeling case. The user-defined material subroutines (VUMAT for laminae and VUMAT for cohesive interface) were used to simulate the impact-induced damage (in terms of indentation depth) by incorporating a moisture concentration curve along the thickness direction obtained from mass diffusion analysis in the VUMAT formulation.

Thirdly, the same damage for all three cases for dry and conditioned specimens were simulated in ABAQUS Standard as a static modeling case. Moisture-dependent viscoelastic user-defined material subroutines (UMAT for laminae and UMAT for cohesive interface) were used to simulate the same damage and its rebound with time (in terms of indentation depth rebound) by incorporating a moisture concentration curve along the thickness direction obtained from mass diffusion analysis in UMAT formulation.

### 3.2.1. Simulation for the Moisture Diffusion Case

Each lamina ply and cohesive interface were modeled as 3D diffusivity (heat transfer) elements (designated as DC3D8 in ABAQUS). The FEM model for specimen with a dimension of 150 mm × 100 mm × 5.022 mm was modeled to simulate the moisture diffusion scenario. The moisture diffusivity of $D_z = 3.64 \times 10^{-7}$ mm$^2$/s was used for the composite specimen (for both laminae ply and cohesive interface). The steady state condition against each applied moisture concentration presented in Table 9 was achieved at approx. 13 months. The moisture diffusion analysis results after a conditioning time of 7200 h for both types of conditioned specimens are shown in Figures 9 and 10. Once the moisture distribution curve through the thickness of the specimen was achieved, as shown in Figure 11, this curve was then incorporated into the VUMAT/UMAT formulation.

**Table 9.** Applied Concentration Data from a typical Psychometric Chart.

| Relative Humidity (% RH) | Temperature (°C) | Pressure (atm, mm of Hg) | Moisture Content (g/m$^3$) | Mass Concentration (ppmv) | Volume (% v) |
|---|---|---|---|---|---|
| 85 | 25 | 1, 760 | 20 | 27430 | 2.74 |
| 100 | 25 | 1, 760 | 23 | 32427 | 3.24 |

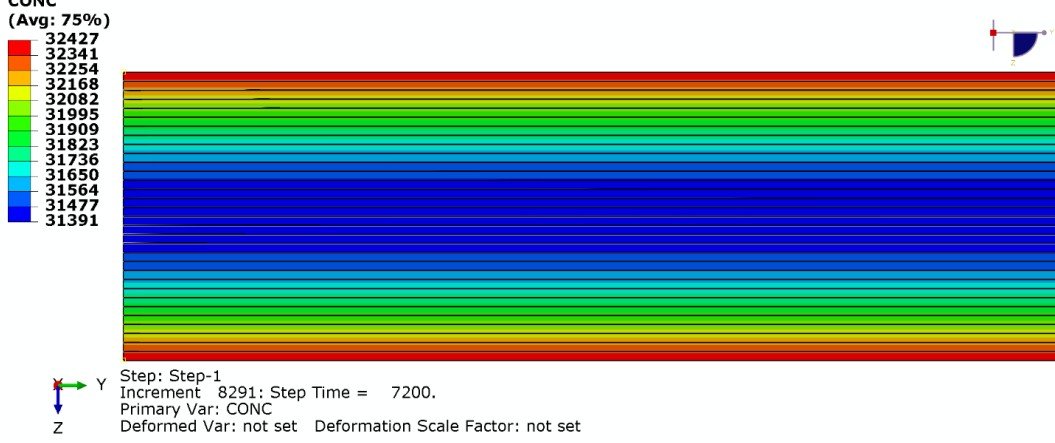

**Figure 9.** Moisture Concentration Diagram after 7200 Hrs (Specimen at 25 °C/RH:100%).

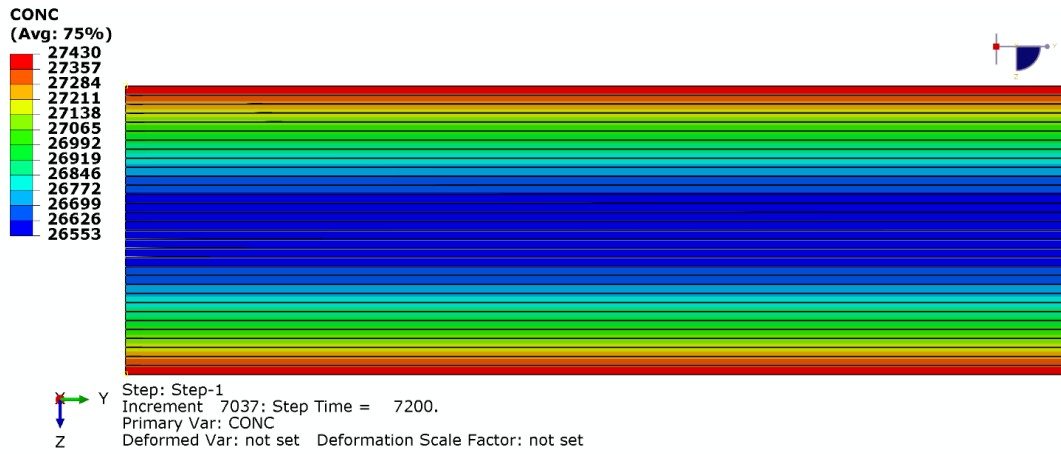

**Figure 10.** Moisture Concentration Diagram after 7200 Hrs (Specimen at 25 °C/RH:85%).

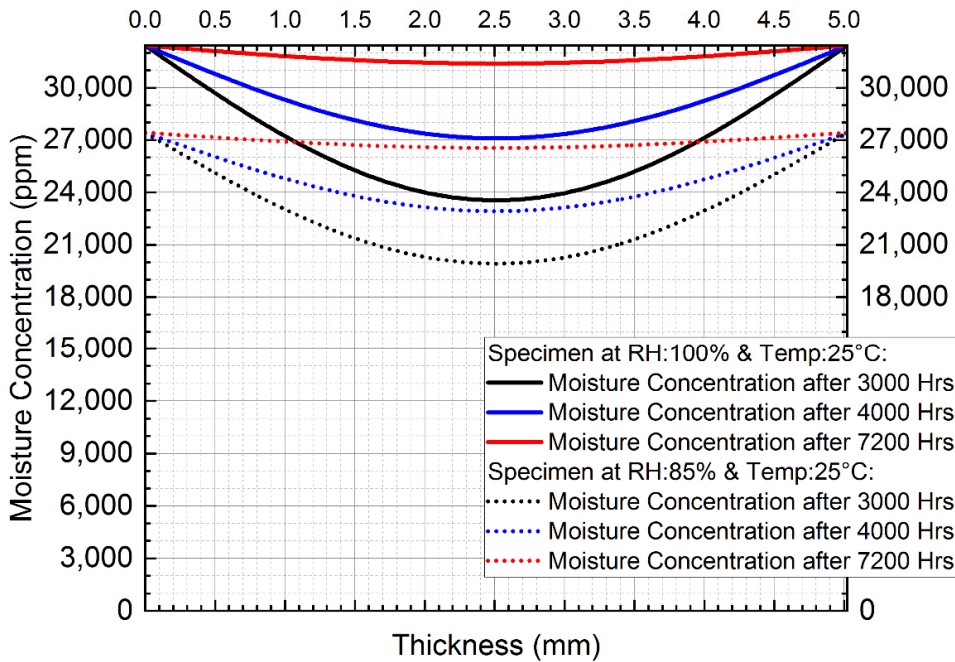

**Figure 11.** Moisture Concentration vs. Thickness Curve.

From LVI testing on a dry specimen, a maximum initial dent depth of 0.348 mm was measured against a 50 J impact energy case. So, in order to optimize the conditioning time, the normalized moisture concentration with respect to the applied moisture concentration at certain thickness values (along the z-direction), i.e., t = 0.471 mm (or t = 4.551 mm), and half of the thickness were considered. The normalized moisture concentration values of 0.921, 0.952, 0.991 and 0.726, 0.836, 0.968 were observed at a thickness of t = 0.471 mm (or t = 4.551 mm) and at one half thickness after conditioning times of 3000, 4000, and 7200 h, respectively, for specimens conditioned at 25 °C/RH:85% and 25 °C/RH:100%. The same findings are shown in Figures 12 and 13.

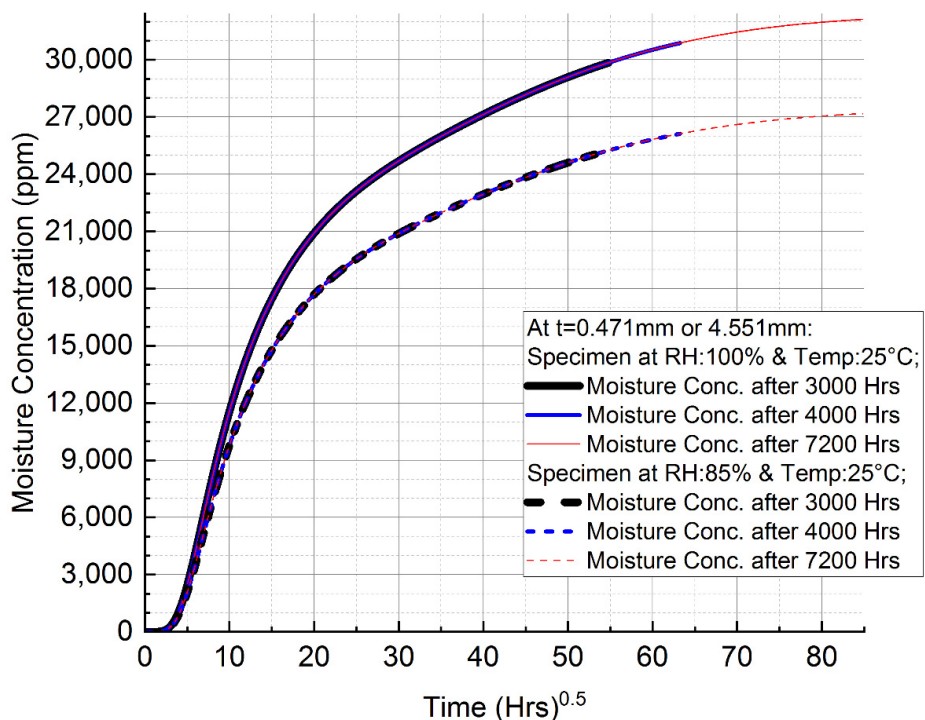

**Figure 12.** Moisture Concentration vs. Time Curve (at thicknesses of t = 0.471 mm or t = 4.551 mm).

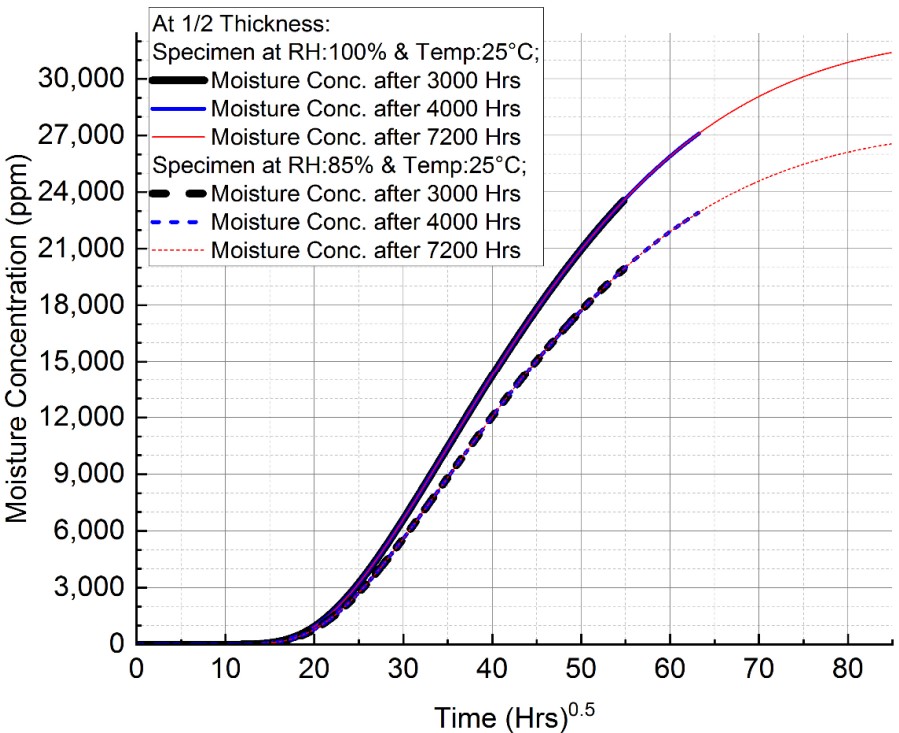

**Figure 13.** Moisture Concentration vs. Time Curve (at 1/2 thickness).

### 3.2.2. Simulation for the Impact Case

The indentation depth was simulated against each impact energy level for dry and conditioned specimens.

To simulate the impact damage in Abaqus Explicit:

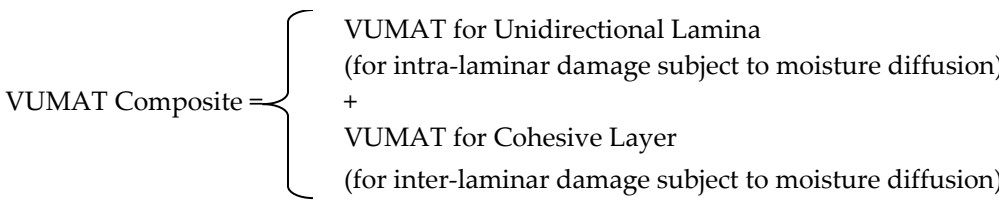

There was a much smaller deformation in the impactor relative to that in the composite laminate. Hence, it was negligible. Therefore, the impactor was modeled as a rigid body during the simulation.

The FEM model depicting the boundary conditions and meshing is shown in Figure 14.

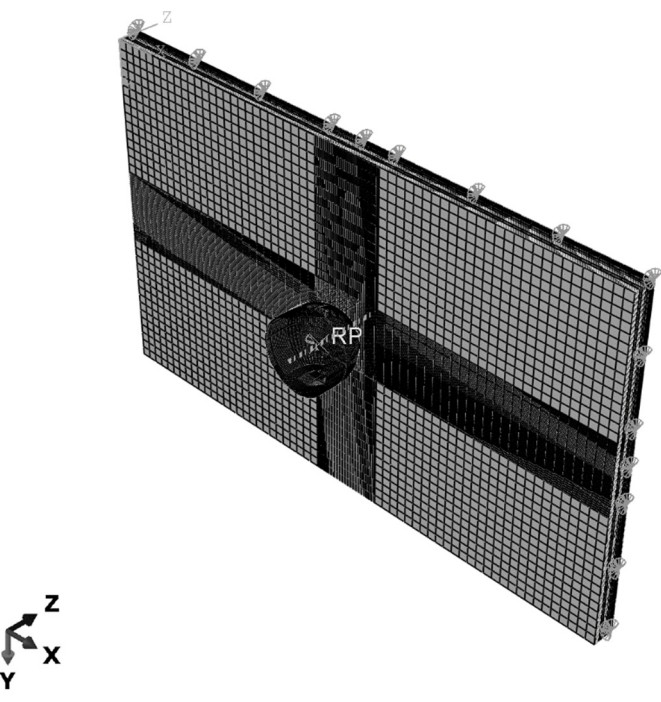

**Figure 14.** The Finite Element Model [48].

The 30, 40, and 50J impact energies were obtained by simulating the impact with the following impactor velocities provided in Table 10.

**Table 10.** Impactor Velocities for different impact energies [48].

|  | Impact Energy (J) | | |
|---|---|---|---|
|  | **50** | **40** | **30** |
| **Velocity (m/s)** | 4.47 | 4.00 | 3.46 |

The values of $\alpha$ (see Equations (19) and (33)) used to incorporate the moisture-dependent variable $(G_0)_{Impact}$ in the VUMAT formulation are presented in Table 11.

**Table 11.** The values of $\alpha$ for $(\mathbf{G}_0)_{Impact}$.

| Hygrothermal Condition | Material Type | $\alpha$ |
|---|---|---|
| 25 °C/RH: 85% | UD Laminae Ply | 0.1420 GPa |
| | Cohesive Interface | 1.3622 GPa/mm |
| 25 °C/RH: 100% | UD Laminae Ply | 0.1880 GPa |
| | Cohesive Interface | 1.8070 GPa/mm |

Each laminae ply was modeled as continuum solid elements (designated as C3D8 in ABAQUS), while the cohesive interface was modeled using element-based cohesive 3D elements (designated as COH3D8 in ABAQUS) with a very small thickness value of $2 \times 10^{-6}$. This increases the overall thickness of the model from 4.96 mm (its original thickness) to 5.022 mm. However, this has negligible effects on the simulation results as damage from all different impact energies only involves a maximum of two cohesive interfaces. For LVI testing, the specimen with a dimension of 150 mm × 100 mm was placed at the fixture base over a hollow slot of dimension 125 mm × 75 mm and secured by 4-toggle clamps in the impact testing machine [53]. The fixture base had guiding pins to restrict displacement in x and y directions. In order to simulate the LVI testing, the specimen with a dimension of 125 mm × 75 mm was modeled only in the FEM with pinned boundary conditions (U1 = U2 = U3 = 0) at the edges. By using pinned boundary conditions at the edges, 12.5 mm from each side along the length and width was simply omitted from the FEM. Thus, simulating the specimen that lies exactly over the hollow slot. The interaction between the plate and the impactor was simulated using the general contact (explicit) regime. The smallest element size in the impact zone was 0.20 mm × 0.20 mm. A penalty friction model was included in the contact property definition and a friction coefficient of 0.3 was used. A semi-automatic mass scaling of $1 \times 10^{-6}$ target time increment was used to reduce the computational time and the element deletion option was imposed.

$\sigma_{33}$ values (stress distribution in the z-direction) for the 50J impact energy for the specimens of three cases, i.e., dry, 85% and 100% relative humidity are shown in Figure 15.

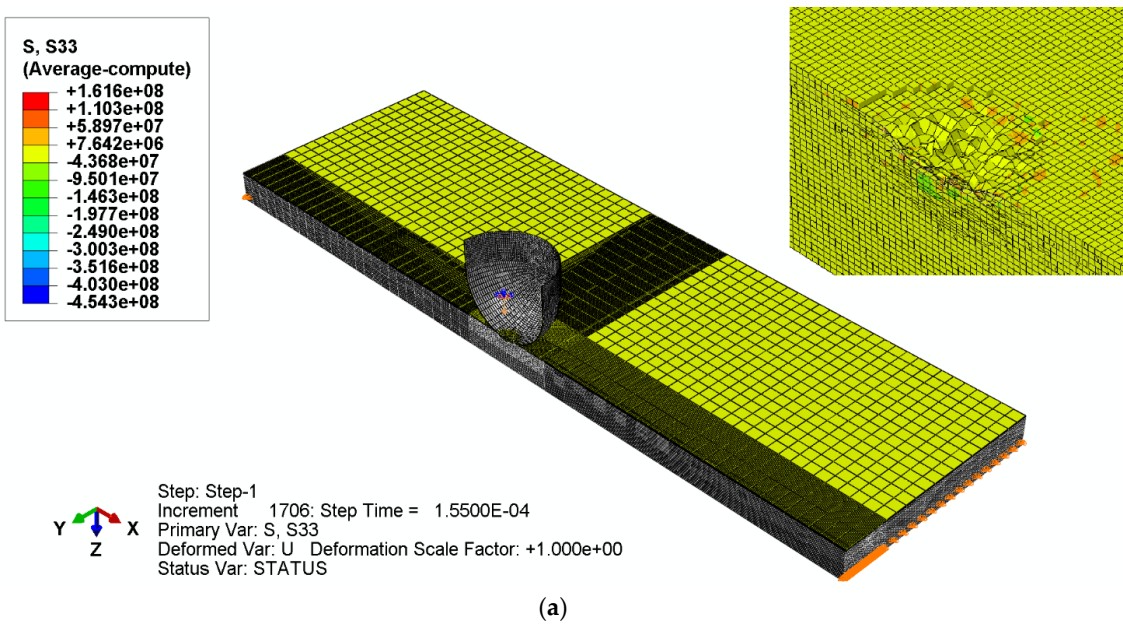

**(a)**

**Figure 15.** *Cont.*

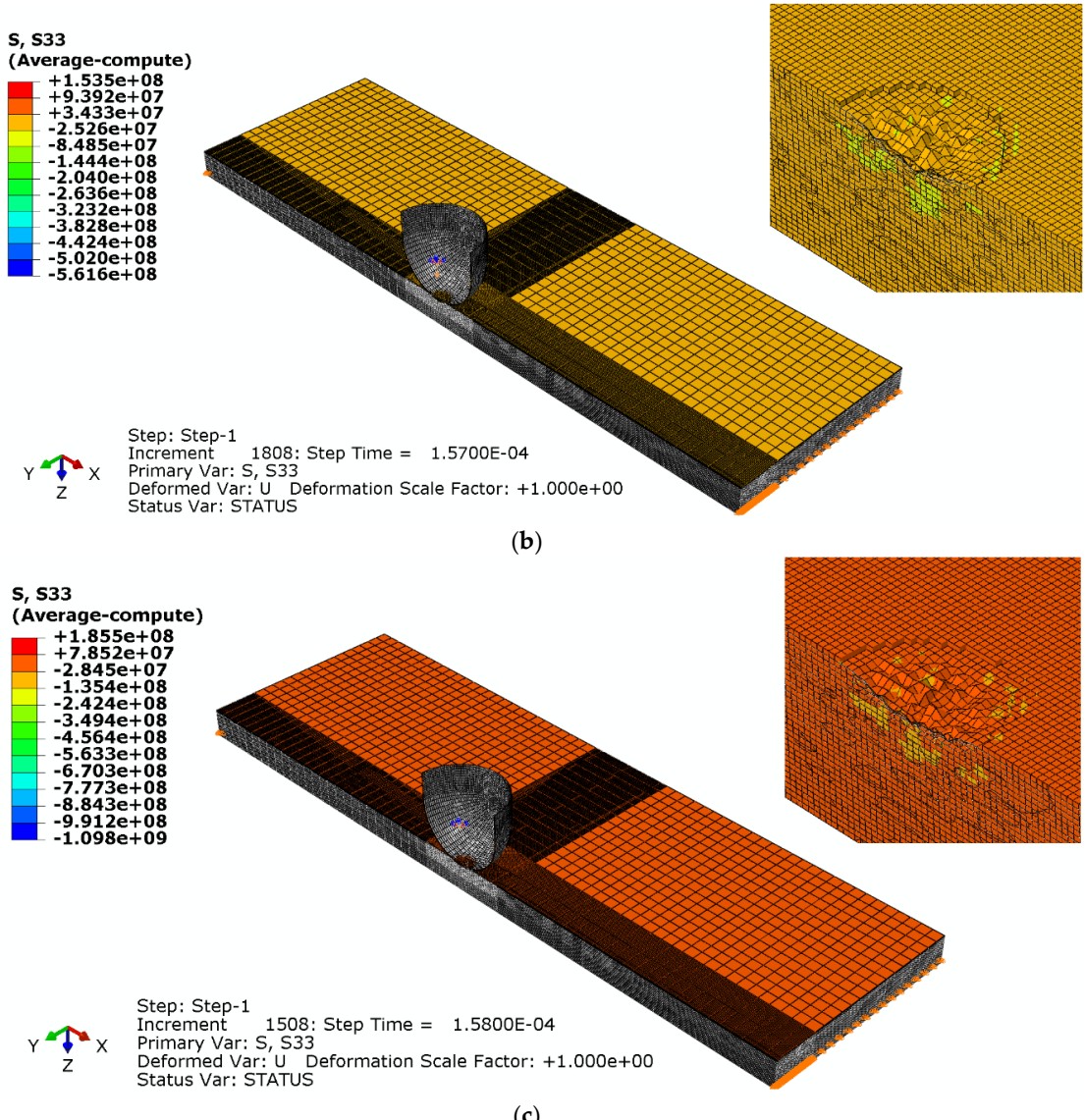

**Figure 15.** $\sigma_{33}$ values (Stress distribution in the z-direction) for 50J Impact Energy Cases. (**a**) Dry Specimen [48]; (**b**) Specimen at RH:85%; (**c**) Specimen at RH:100%.

Similarly, $\sigma_{33}$ values (stress distribution in the z-direction) for 40 and 30J impact energies for the specimens of three cases, i.e., dry, 85% and 100% relative humidity are shown in Figures 16 and 17.

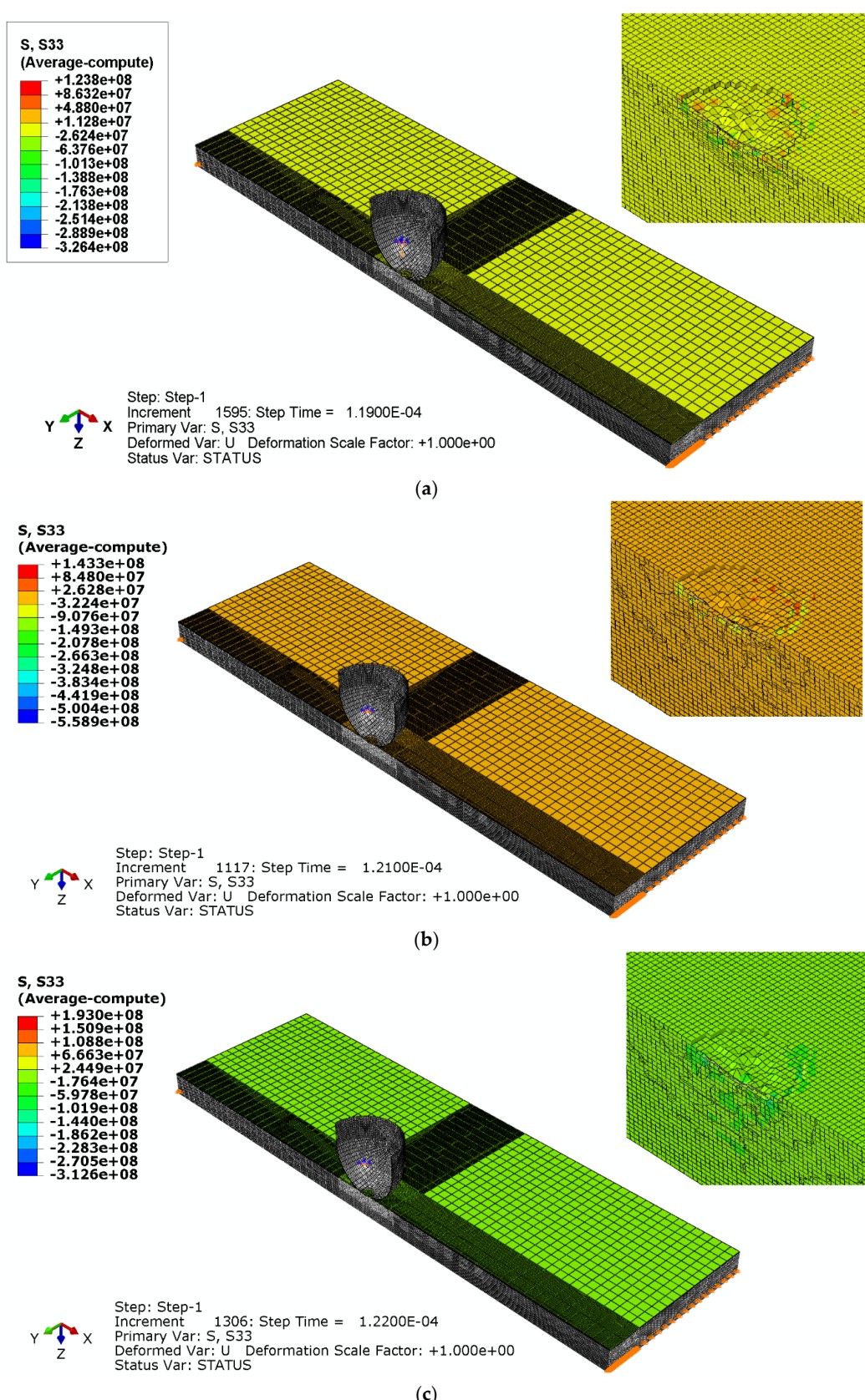

**Figure 16.** $\sigma_{33}$ values (Stress distribution in the z-direction) for 40J Impact Energy Cases. (**a**) Dry Specimen [48]; (**b**) Specimen at RH:85%; (**c**) Specimen at RH:100%.

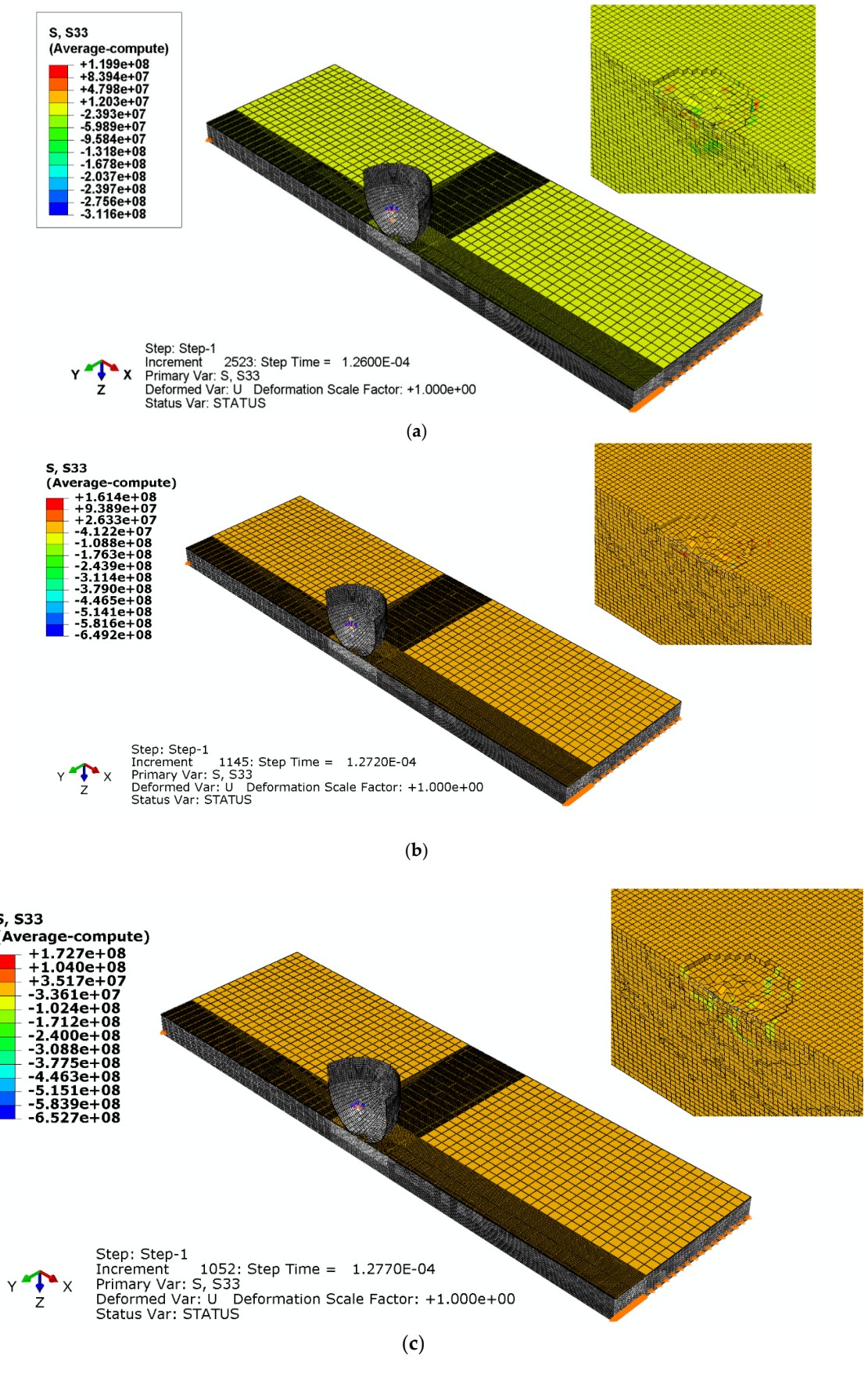

**Figure 17.** $\sigma_{33}$ values (Stress distribution in the z-direction) for 30J Impact Energy Cases. (**a**) Dry Specimen [48]; (**b**) Specimen at RH:85%; (**c**) Specimen at RH:100%.

### 3.2.3. Simulation for the Rebound Case

In the third part of the simulation, indentation depth, which was calculated in the dynamic analysis, was achieved in ABAQUS Standard using a static general analysis. For this, the maximum pressure obtained during dynamic analysis was applied (for a certain period) to obtain the same indentation depth, and then allowed the damage (indentation depth) to evolve/rebound with time due to the viscoelastic behavior of the composite for a dry specimen and due to the moisture-dependent viscoelastic behavior of the composite for a conditioned specimen, respectively (Tables 12–14).

**Table 12.** Summary of both types of simulation for each impact case (dry specimen at 25 °C) [48].

| Impact Case | ABAQUS Explicit Analysis | | | ABAQUS Standard Analysis | |
|---|---|---|---|---|---|
| | Loading Step Time (ms) | Max Pressure (Pa) | Initial Indentation Depth (mm) | Rebound Step Time (h) | Final Indentation Depth (mm) |
| Case-I: 50J | 0.155 | $4.8 \times 10^8$ | 0.348 | 242 | 0.260 |
| Case-II: 40J | 0.119 | $4.4 \times 10^8$ | 0.238 | 244 | 0.204 |
| Case-III: 30J | 0.126 | $4.1 \times 10^8$ | 0.218 | 244.33 | 0.181 |

**Table 13.** Summary of both types of simulation for each impact case (specimen at 25 °C/RH:85%).

| Impact Case | ABAQUS Explicit Analysis | | | ABAQUS Standard Analysis | |
|---|---|---|---|---|---|
| | Loading Step Time (ms) | Max Contact Pressure (Pa) | Initial Indentation Depth (mm) | Rebound Step Time (h) | Final Indentation Depth (mm) |
| Case-I: 50J | 0.157 | $4.5 \times 10^8$ | 0.351 | 280 | 0.217 |
| Case-II: 40J | 0.121 | $4.6 \times 10^8$ | 0.242 | 254 | 0.168 |
| Case-III: 30J | 0.1272 | $3.4 \times 10^8$ | 0.220 | 260 | 0.149 |

**Table 14.** Summary of both types of simulation for each impact case (specimen at 25 °C/RH:100%).

| Impact Case | ABAQUS Explicit Analysis | | | ABAQUS Standard Analysis | |
|---|---|---|---|---|---|
| | Loading Step Time (ms) | Max Contact Pressure (Pa) | Initial Indentation Depth (mm) | Rebound Step Time (h) | Final Indentation Depth (mm) |
| Case-I: 50J | 0.158 | $6.0 \times 10^8$ | 0.353 | 296 | 0.201 |
| Case-II: 40J | 0.122 | $4.9 \times 10^8$ | 0.244 | 272 | 0.156 |
| Case-III: 30J | 0.1277 | $3.7 \times 10^8$ | 0.221 | 271.5 | 0.141 |

To simulate indentation rebound, viscoelasticity subject to moisture diffusion in Abaqus Standard was considered:

$$\text{UMAT Composite} = \left\{ \begin{array}{l} \text{UMAT for Unidirectional Lamina} \\ + \\ \text{UMAT for Cohesive Layer} \end{array} \right.$$

During each impact case for the dry specimen, the specimen absorbs corresponding impact energies. As absorbed energy is different in each case, the rebound is different for each case too. This results in different viscoelastic material behavior of specimens for

each impact case. Therefore, it was deemed necessary to calibrate the viscoelastic material properties of specimens for each impact case separately. Table 15 shows the calibrated viscoelastic material properties for each impact case of the dry specimen for both UD ply and cohesive interface materials. In the 1st Maxwell Chain, elements with relatively smaller elastic stiffness values were chosen over a large period for both material types (UD laminae ply and cohesive interface). However, in the 2nd Maxwell Chain, elements with very large elastic stiffness values were chosen over a relatively smaller period for both material types. From Table 15, it can be seen that it is only necessary to slightly calibrate the 2nd Maxwell Chain for both material types as the absorbed energy in each particular impact case was different, which resulted in corresponding initial dent depths and their rebound over a period of time. Hence, the total dent depth $\Delta d_{Dry}$ was different for different impact energies.

**Table 15.** Viscoelastic Material Properties for each Impact Case [48].

| Impact Case | Material Type | 1st Maxwell Chain | | 2nd Maxwell Chain | |
|---|---|---|---|---|---|
| | | Time 1 (h) | Value 1 (GPa) | Time 2 (h) | Value 2 (GPa) |
| Case-I: 50J | UD Laminae Ply | 240 | 100 | 14 | $2 \times 10^7$ |
| | Cohesive Interface | 240 | *K* Value 1: 55.208 / *G* Value 1: 9.331 | 14 | *K* Value 2: $1.68 \times 10^8$ / *G* Value 2: $2.85 \times 10^7$ |
| Case-II: 40J | UD Laminae Ply | 240 | 100 | 21 | $2 \times 10^7$ |
| | Cohesive Interface | 240 | 55.208 / 9.331 | 21 | $1.93 \times 10^8$ / $3.27 \times 10^7$ |
| Case-III: 30J | UD Laminae Ply | 240 | 100 | 16 | $2 \times 10^7$ |
| | Cohesive Interface | 240 | 55.208 / 9.331 | 16 | $2.15 \times 10^8$ / $3.64 \times 10^7$ |

Viscoelastic material properties, once calibrated for the dry specimen, were used further for predicting the initial dent depth and its rebound over a period of time for the conditioned specimens.

Values of $\alpha$ and $\beta$ (see Equations (12), (13), (26), and (27)) used to incorporate moisture-dependent variables $G_0$ and $G_1$, respectively, in UMAT formulation are presented in Table 16.

**Table 16.** Values of $\alpha$ and $\beta$ for $G_0$ and $G_1$.

| Hygrothermal Condition | Rebound Case | Material Type | $\alpha$ | $\beta$ |
|---|---|---|---|---|
| 25 °C/RH:85% | Case-I: 50J | UD Laminae Ply | The same as in Table 11 | 0.335 |
| | | Cohesive Interface | | |
| | Case-II: 40J | UD Laminae Ply | | 0.580 |
| | | Cohesive Interface | | |
| | Case-III: 30J | UD Laminae Ply | | 0.535 |
| | | Cohesive Interface | | |
| 25 °C/RH:100% | Case-I: 50J | UD Laminae Ply | The same as in Table 11 | 0.400 |
| | | Cohesive Interface | | |
| | Case-II: 40J | UD Laminae Ply | | 0.630 |
| | | Cohesive Interface | | |
| | Case-III: 30J | UD Laminae Ply | | 0.580 |
| | | Cohesive Interface | | |

The results for both types of simulations for different impact energy cases for dry and conditioned specimens are shown in Figures 18–26.

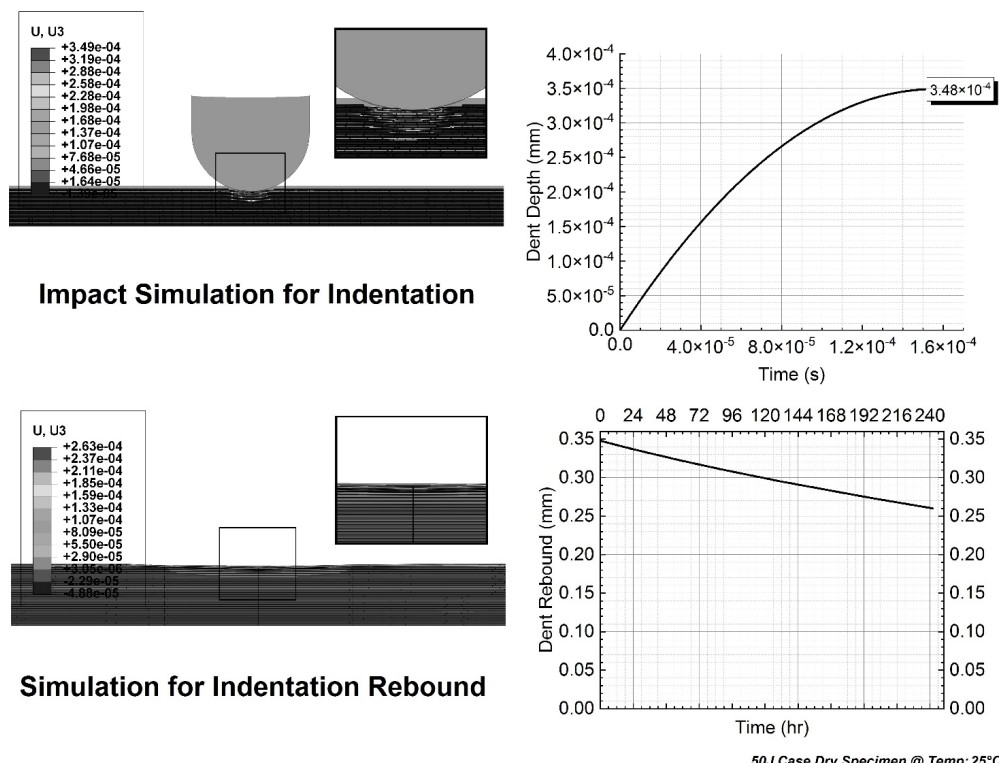

**Figure 18.** Case-I: 50J (Dry Specimen at 25 °C).

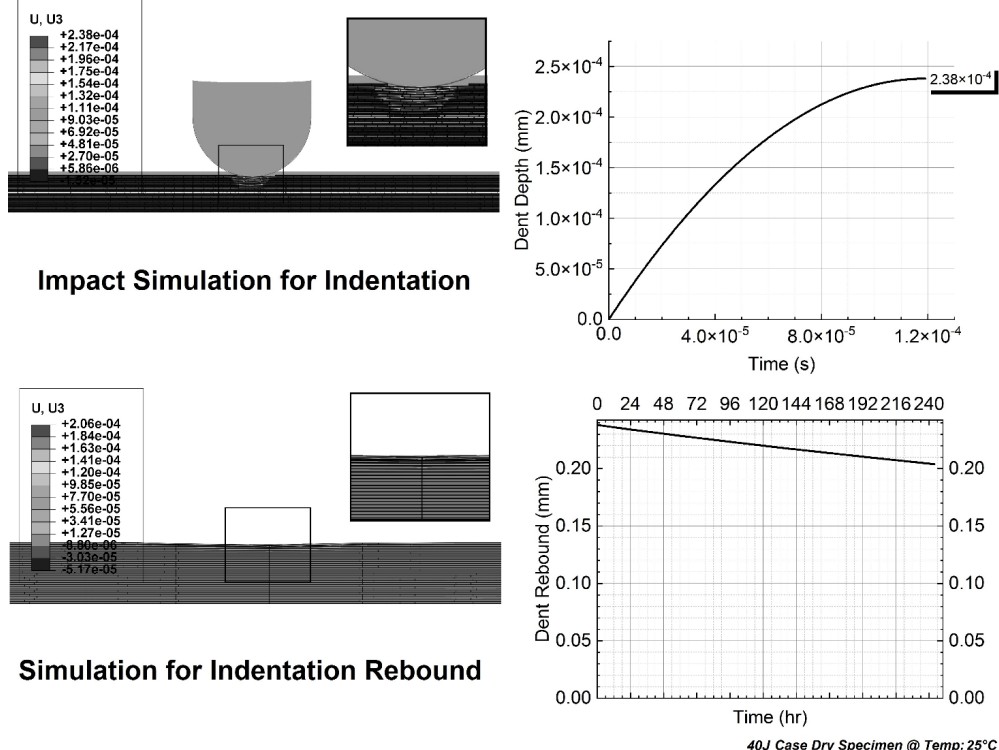

**Figure 19.** Case-II: 40J (Dry Specimen at 25 °C).

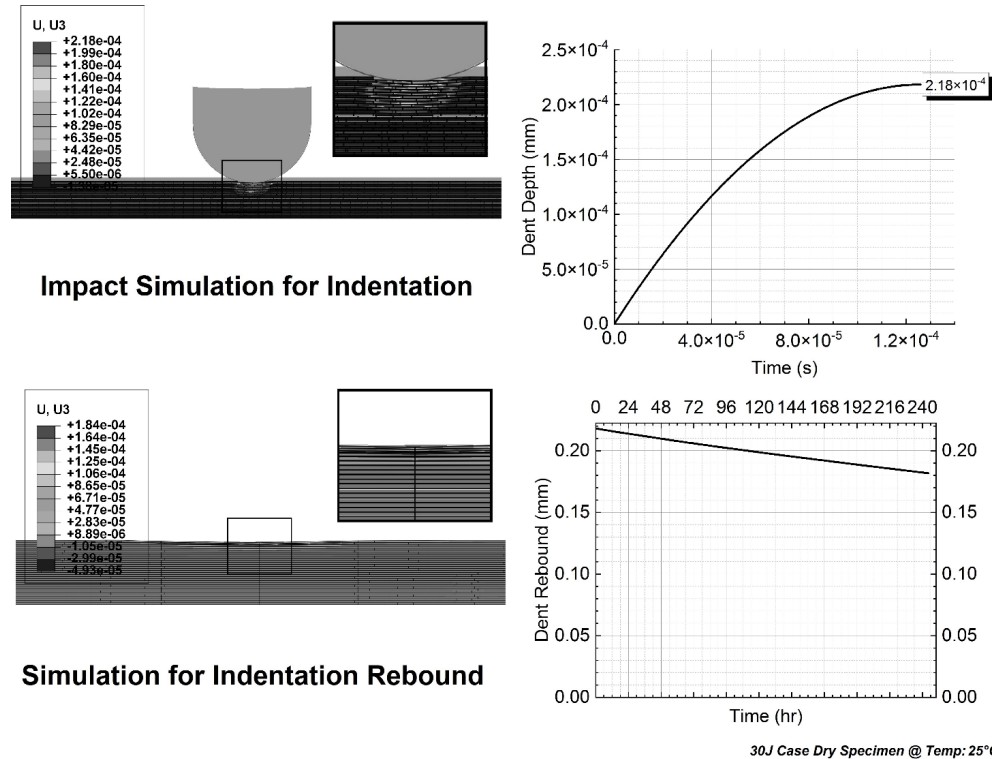

**Figure 20.** Case-III: 30J (Dry Specimen at 25 °C).

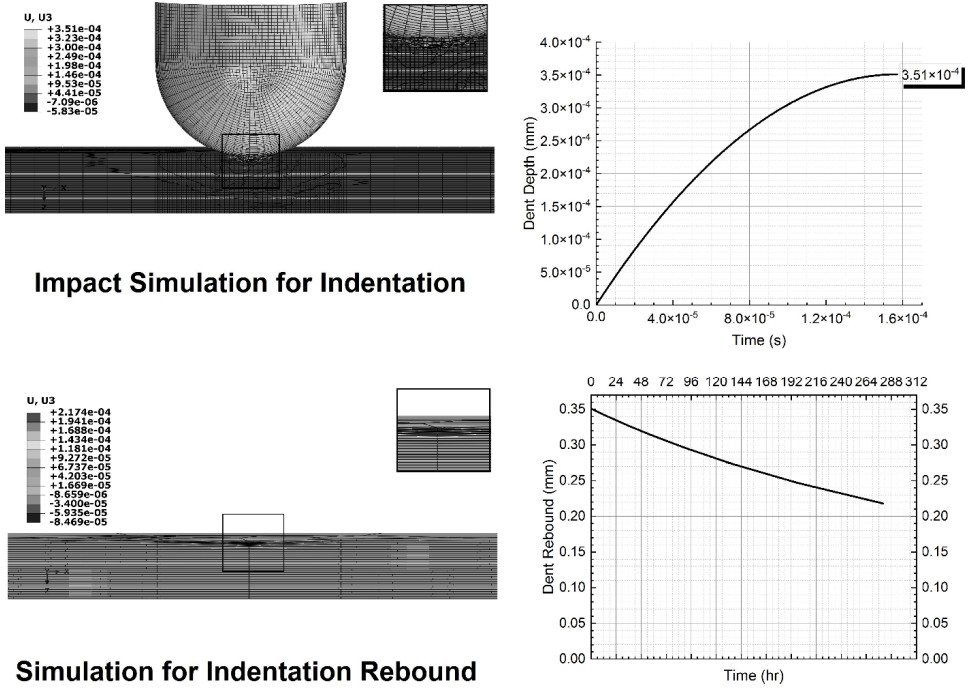

**Figure 21.** Case-I: 50J (Specimen at 25 °C/RH:85%).

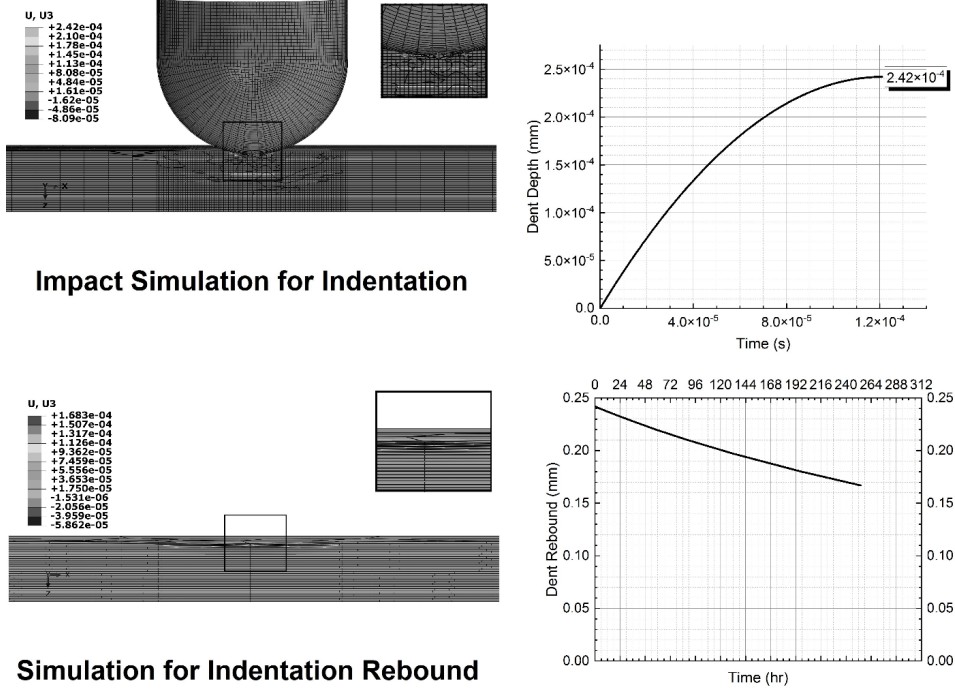

**Figure 22.** Case-II: 40J (Specimen at 25 °C/RH:85%).

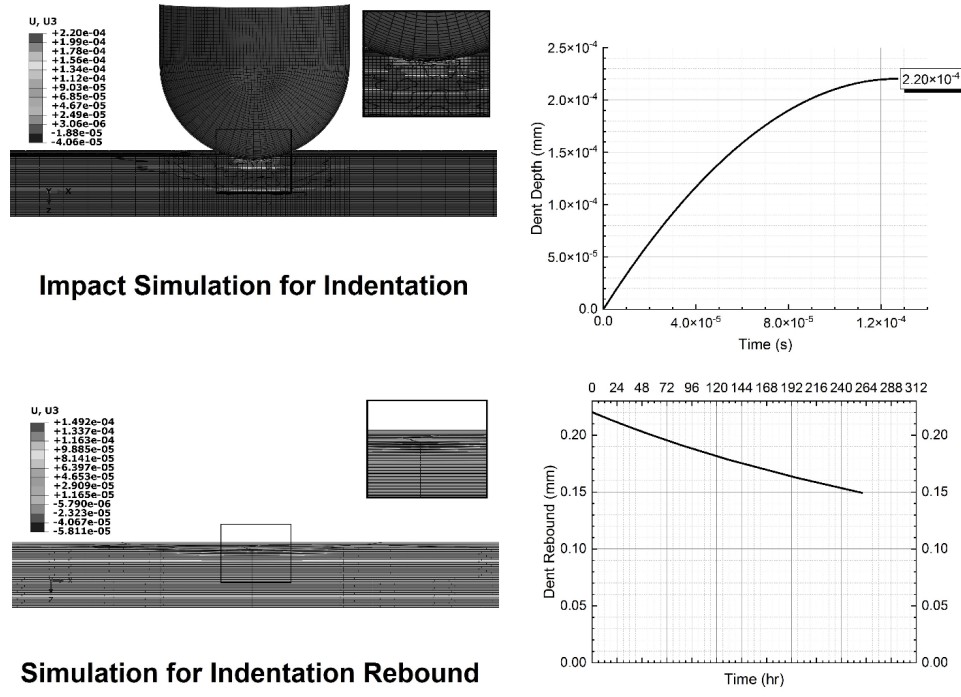

**Figure 23.** Case-III: 30J (Specimen at 25 °C/RH:85%).

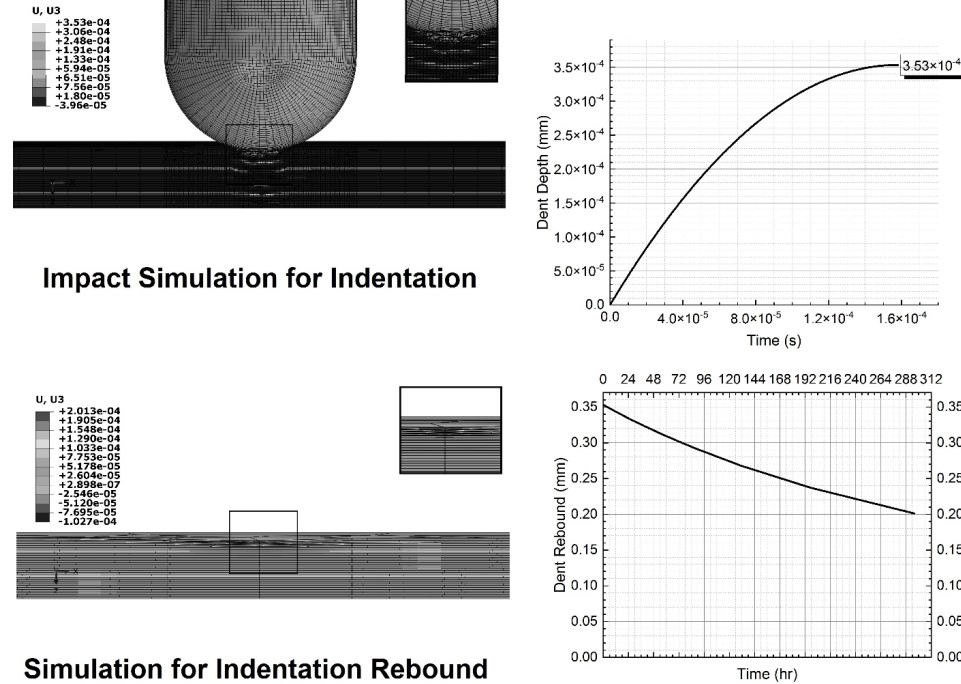

**Figure 24.** Case-I: 50J (Specimen at 25 °C/RH:100%).

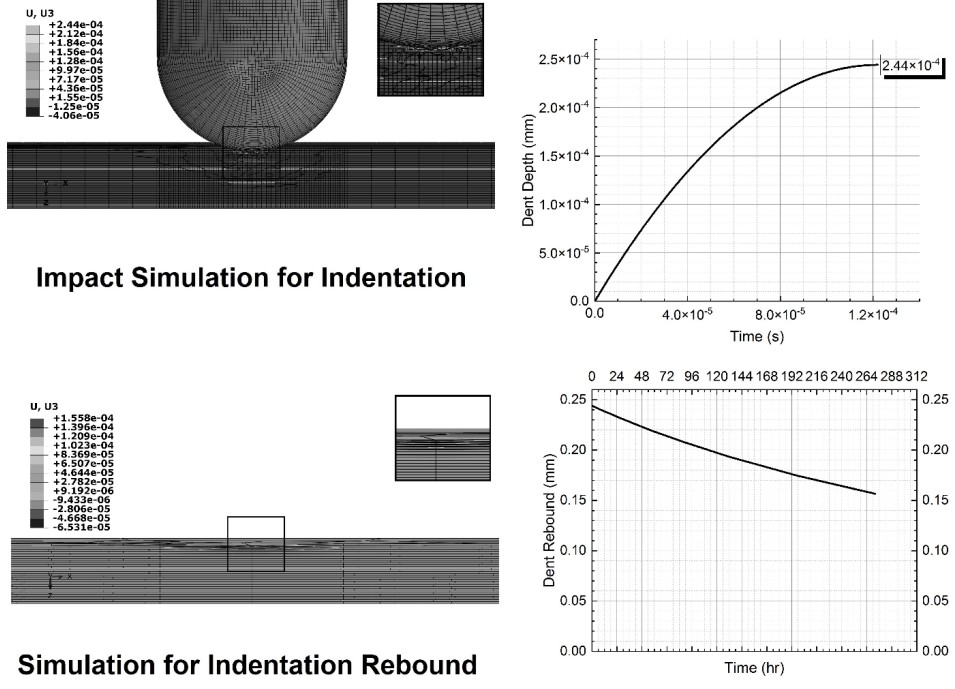

**Figure 25.** Case-II: 40J (Specimen at 25 °C/RH:100%).

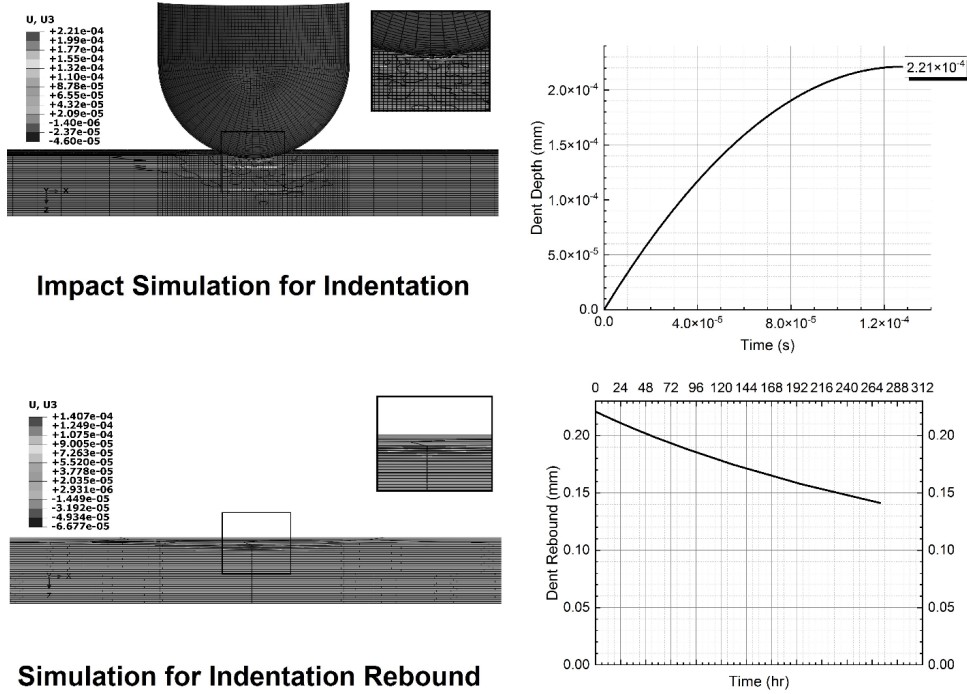

**Figure 26.** Case-III: 30J (Specimen at 25 °C/RH:100%).

Figures 18–26 explain the simulation results obtained from the impact and rebound simulations in terms of initial indentation depth upon impact and its final indentation depth after the rebound over time, respectively. Additionally, the graphs show the dent evolution and rebound with time.

## 4. Results and Discussion

The simulation results agree well with the experimental observations in terms of damage (indentation), depth (initial and final), shape, and size for dry and conditioned specimens. However, the simulation results deviate from the experimental results for the case of the dent rebound path (with a maximum error of 37.90%). This decrease in the original depth of the indentation is faster with time initially after the impact, but it slows down with time and eventually stops. From Figure 27 the predicted dent rebound paths, in spite of being curved, look nearly linear due to the longer rebound time with a very small total dent rebound (in the range of 0.034–0.152 mm). However, the experimental curves are characterized as having a rapid rebound in the early period and then these curves become invariable. This phenomenon is more obvious for the 50 J impact energy cases in dry and conditioned specimens.

The rebound in dent depth is mainly due to the viscoelastic behavior of resin/matrix material in UD laminae ply as well as the viscoelastic behavior of the cohesive interface material in dry specimens. The moisture absorption by the conditioned specimens results in a significant increase in the initial dent depth and a significant decrease in its rebound over a period of time as compared to the same in dry specimens. This is mainly because of the moisture-dependent viscoelastic behavior of resin/matrix materials in UD laminae plies and the moisture-dependent viscoelastic behavior of cohesive interface material.

The graphical representation/comparison of the indentation rebound obtained through experimental data and simulations for all cases is shown in Figure 27.

The simulation results accurately predict the initial and final dent depths, when compared with experimental results, that are provided in Table 17 for the dry specimen.

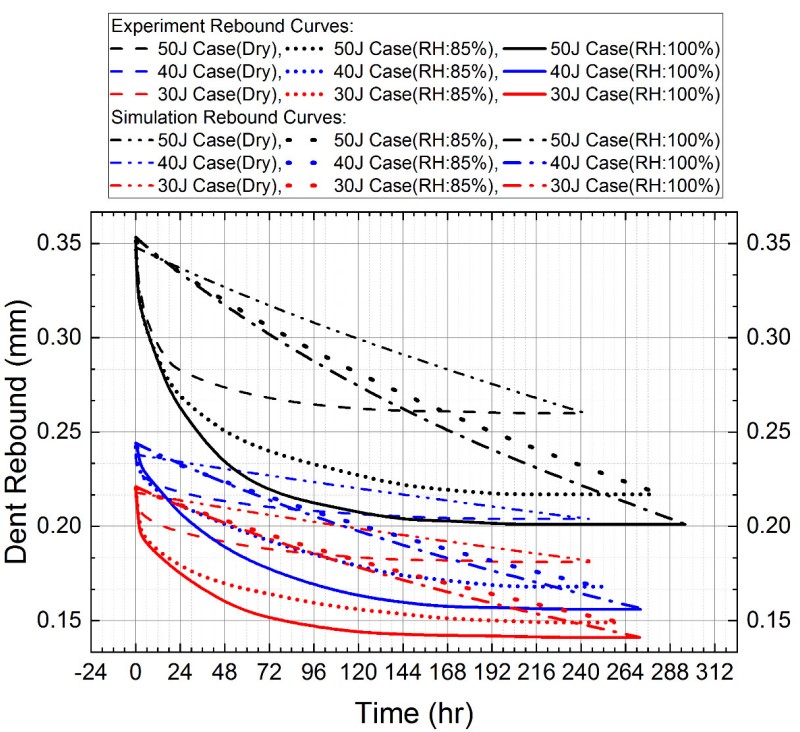

**Figure 27.** Comparison Graph of Experimental and Simulation Indentation Rebound Results.

**Table 17.** Comparison between Experimental and Simulation Results (Dry Specimen at 25 °C) [48].

| Dry Specimen at 25 °C Temperature | | | | | |
|---|---|---|---|---|---|
| **Impact Energy Case** | **Total Dent Rebound** $\Delta d_{Dry}=((d_i)_{Dry}-(d_f)_{Dry})$ **(mm)** | **Way Out** | **Experimental Result** | **Simulation Result** | **Prediction Accuracy** |
| Case-I: 50J | 0.088 | Initial and final dent depths | Matched | Matched | Accurately predicted |
| | | Dent rebound path | The curve is decaying at a faster rate and soon stops decaying before the final point | The curve is decaying at a slower rate and never stops decaying until the final point | Poor prediction Max error: 19.35% |
| Case-II: 40J | 0.034 | Initial and final dent depths | Matched | Matched | Accurately predicted |
| | | Dent rebound path | The curve is decaying at a faster rate and soon stops decaying before the final point | The curve is decaying at a slower rate and never stops decaying until the final point | Fairly inaccurate prediction Max error: 7.97% |
| Case-III: 30J | 0.037 | Initial and final dent depths | Matched | Matched | Accurately predicted |
| | | Dent rebound path | The curve is decaying at a faster rate and soon stops decaying before the final point | The curve is decaying at a slower rate and never stops decaying until the final point | Fairly inaccurate prediction Max error: 9.88% |

Similarly, the simulation results accurately predict initial and final dent depths, when compared with experimental results, that are presented in Tables 18 and 19 for both types of conditioned specimens.

**Table 18.** Comparison between Experimental and Simulation Results (Specimen at 25 °C/RH:85%).

| | | Specimen at 25 °C Temperature and RH: 85% | | | |
|---|---|---|---|---|---|
| **Impact Case** | **Total Dent Rebound** $\Delta d_{RH:85\%}=$ $((d_i)_{RH:85\%}-(d_f)_{RH:85\%})$ **(mm)** | **Way Out** | **Experimental Result** | **Simulation Result** | **Prediction Accuracy** |
| Case-I: 50J | 0.134 | Initial and final dent depths | Matched | Matched | Accurately predicted |
| | | Dent rebound path | Curve is decaying at a faster rate and soon stops decaying before the final point | Curve is decaying at a slower rate and never stops decaying until the final point | Poor prediction Max error: 28.36% |
| Case-II: 40J | 0.074 | Initial and final dent depths | Matched | Matched | Accurately predicted |
| | | Dent rebound path | Curve is decaying at a faster rate and soon stops decaying before the final point | Curve is decaying at a slower rate and never stops decaying until the final point | Fairly inaccurate prediction Max error: 12.29% |
| Case-III: 30J | 0.071 | Initial and final dent depths | Matched | Matched | Accurately predicted |
| | | Dent rebound path | Curve is decaying at a faster rate and soon stops decaying before the final point | Curve is decaying at a slower rate and never stops decaying until the final point | Inaccurate prediction Max error: 20.36% |

**Table 19.** Comparison between Experimental and Simulation Results [Specimen at 25 °C/RH:100%].

| | | Specimen at 25 °C Temperature and RH: 100% | | | |
|---|---|---|---|---|---|
| **Impact Case** | **Total Dent Rebound** $\Delta d_{RH:100\%}=$ $((d_i)_{RH:100\%}-(d_f)_{RH:100\%})$ **(mm)** | **Way Out** | **Experimental Result** | **Simulation Result** | **Prediction Accuracy** |
| Case-I: 50J | 0.152 | Initial and final dent depths | Matched | Matched | Accurately predicted |
| | | Dent rebound path | Curve is decaying at a faster rate and soon stops decaying before the final point | Curve is decaying at a slower rate and never stops decaying until the final point | Poor prediction Max error: 37.90% |
| Case-II: 40J | 0.088 | Initial and final dent depths | Matched | Matched | Accurately predicted |
| | | Dent rebound path | Curve is decaying at a faster rate and soon stops decaying before the final point | Curve is decaying at a slower rate and never stops decaying until the final point | Inaccurate prediction Max error: 21.50% |
| Case-III: 30J | 0.080 | Initial and final dent depths | Matched | Matched | Accurately predicted |
| | | Dent rebound path | Curve is decaying at a faster rate and soon stops decaying before the final point | Curve is decaying at a slower rate and never stops decaying until the final point | Poor prediction Max error: 28.04% |

## 5. Conclusions

In this study, experimental data was acquired to understand the damage incurred and its recovery (the rebound of indentation depth with time) of CFRPs subjected to the low-velocity impact of different impact energies under dry and different hygrothermal conditions with values greater than the damage threshold value, and then the same finite element models were simulated to obtain the results, and the following conclusions were obtained:

Specimens of hygrothermal conditions were found with deeper dents compared with dry ones under the same impact energy, and their rebounds were also more significant. These phenomena were explained as moisture softens epoxy in a composite and elevates its viscosity.

In general, the initial indentation depth was proportional to the impact energy and hygrothermal conditioning. Additionally, the indentation rebound was proportional to this initial indentation depth.

Our simulation results accurately predicted the initial and final dent depths for each impact case. While the decrease in the original depth of indentation remained invariable with time, the rebound path curve obtained via simulation decayed at a slower rate compared to the experimental rebound path curve, adding error to the prediction of the rebound path (with a max. error of 37.90%). This error was proportional to the total dent rebound, which was linked with an increase in the impact energy and relative humidity levels.

For the impact cases of 50, 40, and 30J on specimens at [25 °C/RH:85%] and [25 °C/RH:100%], increases of 0.86%, 1.68%, 0.92% and 1.44%, 2.52%, 1.38% in initial indentation depths were observed, respectively, as compared to the same in the dry specimen at 25 °C. Similarly, for the rebound cases of 50, 40, and 30J on specimens at [25 °C/RH:85%] and [25 °C/RH:100%], decreases of 16.54%, 17.65%, 17.68% and 22.69%, 23.53%, 22.01% in terms of final indentation depths after the rebound over time were observed, respectively, in comparison with the same in the dry specimen at 25 °C.

**Author Contributions:** Conceptualization, M.Y.; methodology, M.Y.; software, M.Y.; validation, M.Y.; formal analysis, M.Y.; investigation, M.Y., Y.Y. and L.W.; resources, C.Z.; data curation, M.Y.; writing—original draft preparation, M.Y.; writing—review and editing, C.Z. and Y.Y.; visualization, M.Y. and L.W.; supervision, C.Z.; project administration, C.Z.; funding acquisition, C.Z. All authors have read and agreed to the published version of the manuscript.

**Funding:** This work was supported by the National Natural Science Foundation of China (11872205), the State Key Laboratory of Mechanics and Control of Mechanical Structures (Nanjing University of Aeronautics and Astronautics, MCMS-E-0221Y02), and the Priority Academic Program Development of Jiangsu Higher Education Institutions.

**Data Availability Statement:** Not applicable.

**Conflicts of Interest:** The authors declare no conflict of interest.

## Abbreviations

| | |
|---|---|
| LVI | Low Velocity Impact |
| PMC | Polymer Matrix Composite |
| B-K | Benzeggagh–Kenane |
| UMAT | User Material Subroutine for ABAQUS/Standard |
| VUMAT | User Material Subroutine for ABAQUS/Explicit |
| CFRP | Carbon Fiber Reinforced Polymer |
| CZM | Cohesive Zone Modeling |
| VCCT | Virtual Crack Closure Technique |
| CDM | Continuum Damage Mechanics |
| UD | Unidirectional |
| GRP | Glass Reinforced Plastic |
| FRC | Fiber Reinforced Concrete |

| TS | Thermoset |
| TP | Thermoplastic |
| AE | Acoustic Emission |
| NDT | Non-Destructive Testing |
| LCA | Life Cycle Analysis |

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
