# Peer review of "Numerical Study of the Hygrothermal Effects on Low Velocity Impact Induced Indentation and Its Rebound in Composite Laminate"

_aerospace, doi:10.3390/aerospace9120802_

Round 1
Reviewer 1 Report
This manuscript treats an interesting topic for the people who are working in the damage behavior of FRP. Some remarks and recommendations are given below for the sake of the paper's improvement.
1) It seems that much research are cited in the introduction to explain the importance of this study. However, the story of the manuscript was not well constructed. The relationship between different references were not given in detail.
2) Equations in Section 2 were not given properly in terms of page organization.
3) CZV , please add long form for the first time in page 2.
4) Page 2: Viscoelastic behavior was mentioned as recoverable behavior. this is not true.
5) The use of Hashin failure criterion for impact damage is doubtful.
6) Figure 4, moisture gain curve x axis generally used (second)^0.5
7) Many numerical studies were carried out and figures of the FEM solutions are given. However, they were not well explained. This case is also similar for experimental studies.
8) The paper was not meticulously written in terms of page format.
Reviewer 2 Report
In the introduction, the authors describe the methods of determining the destruction of composite materials. Among the methods described, there is no information on the use of acoustic emission used to determine the different stages of degradation of materials such as GFRP and CFRP.
Page 5 - after [27] there are two dots, should be only one
Tab. 5-7 - all “depth” values should be recorded with the same accuracy
Fig. 6a - the scale bar is illegible
Fig. 6b - scale bar is missing
Figure 9-10, 15-17 - Using the same scale would make it easier to compare the results shown in the figures
Reviewer 3 Report
This manuscript compares experimental data with the modeled performance of CFRP after Low-velocity impacts. The depth of the resultant dents evaluates the performance, along with the rebound of the depths. This paper addresses an important consideration in the real-world performance of CFRP, that of the hygrothermal conditions. The authors found that their model accurately predicted initial and final dent depths but was inaccurate in predicting the rebound path when compared to the experimental results. The performance of the composite materials was impaired when subjected to elevated moisture due to the moisture dependant visco-elastic behaviour of the composite.
Section 3.1 requires more detail. Where was the CF sourced, what resin is used and what curing procedure was used. Also, there is a typographical error at the top of page 13. I presume the impactor's mass is 5 kg, so remove the ‘0’ from 05. I think an excellent inclusion to this study would be subjecting a second series of dry samples to weathering after the impact event to see how this affects dent rebound. Apart from requiring a few moderate changes to the English, this paper is well-written.
Reviewer 4 Report
The manuscript presents an experimental and numerical study on the moisture effect on the rebound of composite laminates after impact. The manuscript is well written with minor English errors. There are some technical comments that need to be addressed to enhance the manuscript which includes:
-In intorduction, page 2, paragraph "As, CFRP laminates....". This is valid to some extent. There are some limits where the quasi-static indentation can represent LVI experiment. Please define them based on these refs and cite them.
[1] Limits of asymptotic solutions in low-velocity impact of
composite plates. Compos Struct 2007;81(4):568–74.
[2]Scaling effects of composite laminates under out-of-plane loading. Composites Part A: Applied Science and Manufacturing, 2019, 116, 1-12.
- In the introduction, page 2, "Thomas [29] in his paper......", there are other recent works that is easy to check and discussed this phenomenon in a detailed way as ref [3].
[3] A quasi-static indentation test to elucidate the
sequence of damage events in low velocity impacts on composite laminates.
Compos Part A: Appl Sci Manuf 2016;82:180–9.
- In section 3.2.1, how did you calculate the moisture diffusivity?
- Figs. 15, 16, 17 doesn't show the distributions. To highligh this, please show the deformed shape without mesh.
Also, i recommend to show the distribution at single energy but for the three cases, dry, 85 and 100 humidity. So, it will be easy to understand and follow the difference.
- In section 4, another parameter that affect a lot the rebound is the level of damage inside the laminate. So, two factors here affect the rebound, the viscosity of the material and the level of damage. this explains the different behaviour of the relaxation curve (fig. 27), because if you consider only viscoelasticity, the damage level should not have effect if you normalized the results.
-Conclusions need to be shortened.
Round 2
Reviewer 1 Report
The authors have carefully modified and improved their manuscript. I recommend it publication. However, the change the auhors did in the introduction as giving many sub-sections (sub-headings) is not common. I recommend you to eliminate them and arrange the transitions between paragraphs more natural in order not to disturb the flow of the introduction.
Reviewer 4 Report
The authors addressed the comments adequately.
